# Zero-Shot Learning with Common Sense Knowledge Graphs

**Nihal V. Nayak**                                                   *nnayak2@cs.brown.edu*
*Department of Computer Science*
*Brown University*

**Stephen H. Bach**                                                   *sbach@cs.brown.edu*
*Department of Computer Science*
*Brown University*

**Reviewed on OpenReview:** *https://openreview.net/forum?id=h1zuM6cXpH*

## Abstract

Zero-shot learning relies on semantic class representations such as hand-engineered attributes or learned embeddings to predict classes without any labeled examples. We propose to learn class representations by embedding nodes from common sense knowledge graphs in a vector space. Common sense knowledge graphs are an untapped source of explicit high-level knowledge that requires little human effort to apply to a range of tasks. To capture the knowledge in the graph, we introduce ZSL-KG, a general-purpose framework with a novel transformer graph convolutional network (TrGCN) for generating class representations. Our proposed TrGCN architecture computes non-linear combinations of node neighbourhoods. Our results show that ZSL-KG improves over existing WordNet-based methods on five out of six zero-shot benchmark datasets in language and vision. The code is available at https://github.com/BatsResearch/zsl-kg.

## 1 Introduction

Zero-shot learning is a training strategy which allows a machine learning model to predict novel classes without the need for any labeled examples for the new classes (Romera-Paredes & Torr, 2015; Socher et al., 2013; Wang et al., 2019). These models are trained on a set of labeled examples from *seen* classes, along with their class representations. During inference, new class representations are provided for the *unseen* classes. Previous zero-shot learning systems have used hand-engineered attributes (Farhadi et al., 2009; Lampert et al., 2013), pretrained embeddings (Frome et al., 2013) and learned embeddings (e.g., sentence embeddings) (Reed et al., 2016) as class representations.

Previous approaches for class representations have various limitations. Attribute-based methods provide rich features and have achieved state-of-the-art results on several zero-shot object classification datasets, but the attributes have to be fixed ahead of time for the unseen classes and cannot adapt to new classes beyond the dataset. Furthermore, creating attribute datasets can take up to thousands of hours of labor (Zhao et al., 2019). Finally, attribute-based methods may not be readily applicable to tasks in language, as they might require greater nuance and flexibility (Gupta et al., 2020). Alternatively, pretrained embeddings such as GloVe (Pennington et al., 2014) and Word2Vec (Mikolov et al., 2013) offer the flexibility of easily adapting to new classes but rely on unsupervised training on large corpora—which may not provide distinguishing characteristics necessary for zero-shot learning. Many methods lie within this spectrum and learn class representations for zero-shot tasks from descriptions such as text and image prototypes.

Recently, there is growing interest in methods using graph neural networks on the ImageNet graph, a noun subset of the WordNet graph, to learn to map nodes to class representations (Wang et al., 2018). These graph-based methods have achieved strong performance on zero-shot object classification. They offer the benefits of high-level knowledge from the graph, with the flexibility of pre-trained embeddings. They are general-purpose and applicable to a broader range of tasks beyond object classification, since we show

that they can be adapted to language datasets as well. However, the ImageNet graph is specialized to an object-type hierarchy, and may not provide rich features as suitable for a wide range of downstream tasks in multiple domains.

In our work, we propose to learn class representations better suited for a wider range of tasks from common sense knowledge graphs. Common sense knowledge graphs (Liu & Singh, 2004; Speer et al., 2017; Zhang et al., 2020; Ilievski et al., 2020) are an untapped source of explicit high-level knowledge that requires little human effort to apply to a range of tasks. These graphs have explicit edges between related concept nodes and provide valuable information to distinguish between different concepts. However, adapting existing zero-shot learning frameworks to learn class representations from common sense knowledge graphs is challenging in several ways. GCNZ (Wang et al., 2018) learns graph neural networks with a symmetrically normalized graph Laplacian and requires the entire graph structure during training, i.e., GCNZ is not inductive. Common sense knowledge graphs can be large (2 million to 21 million edges) and training a graph neural network on the entire graph can be prohibitively expensive. DGP (Kampffmeyer et al., 2019) is an inductive method and aims to generate expressive class representations, but assumes a directed acyclic graph such as WordNet. Common sense knowledge graphs do not have a directed acyclic graph structure.

To address these limitations, we propose ZSL-KG, a framework with a novel transformer graph convolutional network (TrGCN) to learn class representations. Graph neural networks learn to represent the structure of graphs by aggregating information from each node's neighbourhood. Aggregation techniques used in GCNZ, DGP, and most other graph neural network approaches are linear, in the sense that they take a (possibly weighted) mean or maximum of the neighbourhood features. To capture the complex information in the common sense knowledge graph, TrGCN learns a transformer-based aggregator to compute a non-linear combination of the node neighbours and increases the expressiveness of the class representation (Hamilton et al., 2017a). Our framework is also inductive, i.e., the graph neural network can be executed on graphs that are different from the training graph, which is necessary for generalized zero-shot learning under which the test classes are unknown during training.

Our main contributions are the following:

1. We propose to learn to map nodes in common sense knowledge graphs to class representations for zero-shot learning.
2. We present ZSL-KG, a framework based on graph neural networks with a novel transformer graph convolutional network (TrGCN). Our proposed architecture learns permutation invariant non-linear combinations of the nodes' neighbourhoods and generates expressive class representations.
3. We demonstrate that graph-based zero-shot learning frameworks—originally designed for object classification—are also applicable to language tasks.
4. ZSL-KG achieves new state-of-the-art accuracies for zero-shot learning on the OntoNotes (Gillick et al., 2014), BBN (Weischedel & Brunstein, 2005), attribute Pascal Yahoo (aPY) (Farhadi et al., 2009) and SNIPS-NLU (Coucke et al., 2018) datasets. Finally, ZSL-KG also outperforms existing WordNet-based zero-shot learning frameworks on the ImageNet 22K dataset (all classes used for testing) (Deng et al., 2009), while maintaining competitive performance on the Animals with Attributes 2 (AWA2) dataset (Xian et al., 2018a).

## 2 Related Work

We broadly describe related works on zero-shot learning, graph neural networks, and common sense knowledge graphs.

**Zero-Shot Learning.** Zero-shot learning has received increased interest over the years in both language and vision (Farhadi et al., 2009; Brown et al., 2020). In our work, we are focused on zero-shot fine-grained entity typing, intent classification, and object classification. Zero-shot fine-grained entity typing has been previously studied with a variety of class representations (Yogatama et al., 2015; Yuan & Downey, 2018; Obeidat et al., 2019). ZOE (Zhou et al., 2018) is a specialized method for zero-shot fine-grained entity typing and shows accuracy competitive with supervised methods. It uses hand-crafted type definitions for each dataset and tuned thresholds, developed with knowledge of the unseen classes, which makes them a transductive zero-shot

method. Our work focuses on graph-based methods in zero-shot learning where unseen types are not revealed during training or tuning. There has been much work on zero-shot object classification (Frome et al., 2013; Lampert et al., 2013; Wang et al., 2018; Xian et al., 2018b). Recent works in zero-shot learning have used graph neural networks for object classification (Wang et al., 2018; Kampffmeyer et al., 2019). In our work, we extend their approach to common sense knowledge graphs to generate class representations with a novel transformer graph convolutional network. Liu et al. (2020a) proposed a graph propagation mechanism for zero-shot object classification. However, they construct the graph by leveraging the predefined attributes for the classes, which are not easily adapted to language tasks. HVE (Liu et al., 2020b) learns similarity between the image representation and the class representations in the hyperbolic space. However, the class representations are not learned with the task instead they are pretrained hyperbolic embeddings for GloVe and WordNet whereas, in our work, we focus on methods that learn class representations explicitly from the knowledge graphs in the task. More recent work on zero-shot object classification (Chen et al., 2021; 2022a;b) use transformer-based networks with attributes whereas we learn a transformer graph convolutional networks with a common sense knowledge graph and outperform them on both AWA2 and aPY datasets. Other notable works in zero-shot learning include text classification (Chang et al., 2008; Yin et al., 2019), video action recognition (Gan et al., 2015), machine translation (Johnson et al., 2017), and more (Wang et al., 2019).

Recent work shows that large-scale models like CLIP (Radford et al., 2021) and GPT-3 (Brown et al., 2020) exhibit zero-shot generalization to new tasks and datasets. However, a key challenge is the potential overlap of the seen and unseen datasets. For example, analysis of the training data used in CLIP showed that 24 out of the 35 held-out datasets detected overlap (Radford et al., 2021). In contrast, we systematically study the zero-shot generalization of classes without any overlap in the seen and unseen classes. Furthermore, these models use text-based prompts to represent classes that may be not provide sufficient control to represent complex fine-grained classes. In contrast, ZSL-KG offers a flexible way of representing unseen fine-grained and complex classes in a knowledge graph. One limitation to note is that ZSL-KG requires the classes to be mapped to the knowledge graph. Fine-grained object classification datasets may contain classes that might be missing from the off-the-shelf ConeptNet graph. For example, ConceptNet does not have a node for the class Forster's Tern in the CUB dataset (Wah et al., 2011). To extend ZSL-KG to rare and domain-specific classes, we would need a specialized knowledge graph.

**Graph Neural Networks.** Graph neural networks learn node embeddings that reflect the structure of the graph (Hamilton et al., 2017b). Recent work on graph neural networks has demonstrated significant improvements for several downstream tasks such as node classification and graph classification (Hamilton et al., 2017a; Kipf & Welling, 2017; Marcheggiani & Titov, 2017; Schlichtkrull et al., 2018; Veličković et al., 2018; Wu et al., 2019; Shang et al., 2019; Vashishth et al., 2020). In this work, we introduce transformer graph convolutional networks for zero-shot learning. Since our preprint appeared on ArXiv, several variants of graph transformers have been proposed in the literature (Dwivedi & Bresson, 2021; Kreuzer et al., 2021; Ying et al., 2021; Mialon et al., 2021; Dwivedi et al., 2022). The primary motivation of their work is to model long-range interactions in the graph. They assume all the nodes are connected to each other and learn a transformer with positional and structural representations over the entire graph. However, this increases the computational complexity of the model. Suppose we have a graph with $n$ nodes, traditional graph neural networks have a complexity of $O(n)$ whereas graph transformers have a complexity of $O(n^2)$ to compute embeddings for all the nodes in the graph. In contrast, our proposed TrGCN increases the expressivity of the node representations by operating on the local neighbourhood with a complexity of $O(m^2 \cdot n)$ where $m << n$ is the maximum number of node neighbours during training and testing. Prior work (Hu et al., 2020; Yun et al., 2020) has also considered transformers as a method to learn meta-paths in heterogeneous graphs rather than as a neighbourhood aggregation technique. Finally, several diverse applications using graph neural networks have been explored: fine-grained entity typing (Xiong et al., 2019), text classification (Yao et al., 2019), reinforcement learning (Adhikari et al., 2020) and neural machine translation (Bastings et al., 2017). For a more in-depth review, we point readers to Wu et al. (2021).

**Common Sense Knowledge Graphs.** Common sense knowledge graphs have been applied to a range of tasks (Lin et al., 2019; Zhang et al., 2019b; Bhagavatula et al., 2019; Bosselut et al., 2019; Shwartz et al., 2020; Yasunaga et al., 2021). ConceptNet has been used for transductive zero-shot text classification as shallow

features for class representation (Zhang et al., 2019b) along with other knowledge sources such as pretrained emebeddings and textual description, which differs from our work as we generate dense vector representation from ConceptNet with TrGCN. Finally, TGG (Zhang et al., 2019a) uses common sense knowledge graph and graph neural networks for transductive zero-shot object classification. TGG learns to model seen-unseen relations with a graph neural network. Their framework is transductive as they require knowledge of unseen classes during training and use hand-crafted attributes. In contrast, ZSL-KG is an inductive framework which does not require explicit knowledge of the unseen classes during training and learn class representations from the common sense knowledge graph.

## 3    Background

In this section, we briefly summarize zero-shot learning and graph neural networks.

**Zero-Shot Learning.** Zero-shot learning has several variations (Wang et al., 2019). Formally, we have the training set $\mathbb{S} = \{(x_1, y_1), ..., (x_n, y_n)\}$ where $y_i$ belongs to the set of seen classes $Y_S$. In the conventional zero-shot learning setting (ZSL), we assign examples to the correct class(es) from the unseen classes $Y_U$ and $Y_S \cap Y_U = \varnothing$. In generalized zero-shot learning (GZSL) setting, we assign examples to the correct class(es) from the set of seen and unseen classes $Y_{U+S} = Y_S \cup Y_U$. Based on prior work in each task, we choose to evaluate in conventional zero-shot learning, generalized zero-shot learning, or both.

Zero-shot classifiers are trained on the seen classes, but unlike traditional supervised learning, they are trained along with class representations such as attributes, pretrained embeddings, etc. Recent approaches learn a class encoder $\phi(y) \in \mathbb{R}^d$ to produce vector-valued class representations from an initial input, such as a string or other identifier of the class. (In our case, $y$ is a node in a graph and its $k$-hop neighborhood.) During inference, the class representations are used to label examples with the unseen classes by passing the examples through an example encoder $\theta(x) \in \mathbb{R}^d$ and predicting the class whose representation has the highest inner product with the example representation.

Recent work in zero-shot learning commonly uses one of two approaches to learn the class encoder $\phi(y)$. One approach uses a bilinear similarity function defined by a compatibility matrix $\boldsymbol{W} \in \mathbb{R}^{d \times d}$ (Frome et al., 2013; Xian et al., 2018b):

$$f\left(\theta(x), \boldsymbol{W}, \phi(y)\right) = \theta(x)^{\mathrm{T}} \boldsymbol{W} \phi(y) . \tag{1}$$

The bilinear similarity function gives a score for each example-class pair. The parameters of $\theta$, $\boldsymbol{W}$, and $\phi$ are learned by taking a softmax over $f$ for all possible seen classes $y \in Y_S$ and minimizing either the cross entropy loss or a ranking loss with respect to the true labels. In other words, $f$ should give a higher score for the correct class(es) and lower scores for the incorrect classes. $\boldsymbol{W}$ is often constrained to be low rank, to reduce the number of learnable parameters (Yogatama et al., 2015). Lastly, other variants of the similarity function add minor variations such as non-linearities between factors of $\boldsymbol{W}$ (Xian et al., 2016).

The other common approach is to first train a neural network classifier in a supervised fashion. The final fully connected layer of this network has a vector representation for each seen class, and the remaining layers are used as the example encoder $\theta(x)$. Then, the class encoder $\phi(y)$ is trained by minimizing the L2 loss between the representations from supervised learning and $\phi(y)$ (Socher et al., 2013; Wang et al., 2018). The class encoder that we propose in Section 4 can be plugged into either approach.

**Graph Neural Networks.** The basic idea behind graph neural networks is to learn node embeddings that reflect the structure of the graph (Hamilton et al., 2017b). Consider the graph $\mathcal{G} = (V, E, R)$, where $V$ is the set of vertices with node features $X_v$ and $(v_i, r, v_j) \in E$ are the labeled edges and $r \in R$ are the relation types. Graph neural networks learn node embeddings by iterative aggregation of the k-hop neighbourhood. Each layer of a graph neural network has two main components AGGREGATE and COMBINE (Xu et al., 2019):

$$\boldsymbol{a}_v^{(l)} = \mathrm{AGGREGATE}^{(l)} \left(\left\{\boldsymbol{h}_u^{(l-1)} \; \forall u \in \mathcal{N}(v)\right\}\right) \tag{2}$$

where $\boldsymbol{a}_v^{(l)} \in \mathbb{R}^{d_{l-1}}$ is the aggregated node feature of the neighbourhood, $\boldsymbol{h}_u^{(l-1)}$ is the node feature in neighbourhood $\mathcal{N}(.)$ of node $v$ including a self loop. The aggregated node is passed to the COMBINE to

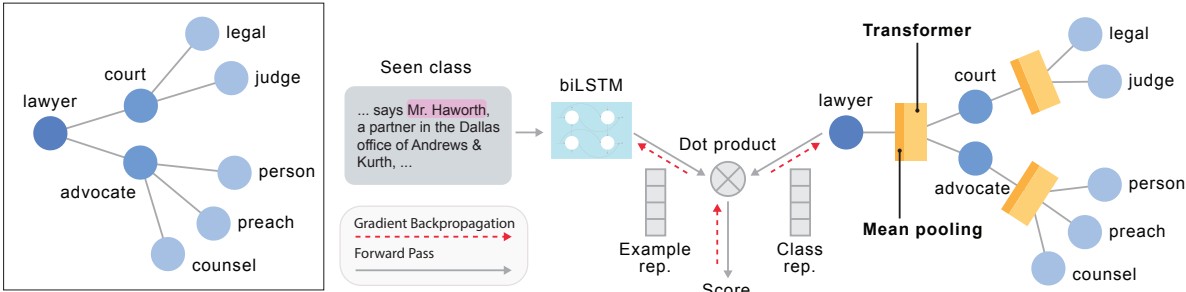

Figure 1: *Left:*A sample from the 2-hop neighbourhood for the concept `lawyer` from ConceptNet. *Right:* The ZSL-KG architecture. The text with the named entity is passed through the example encoder (biLSTM with attention) to generate the example representation. For the class representation, we take the k-hop neighbourhood and pass the nodes through their hop-specific transformer graph convolutional network (TrGCN). Next, we take the dot products between the example representations and the class representations to get compatibility scores. Finally, we train the example encoder and the class encoder by minimizing the task-specific loss over the true labels.

generate the node representation $\boldsymbol{h}_v^{(l)} \in \mathbb{R}^{d_l}$ for the $l$-th layer:

$$\boldsymbol{h}_v^{(l)} = \text{COMBINE}^{(l)}\left(\boldsymbol{h}_v^{(l-1)}, \boldsymbol{a}_v^{(l)}\right) \tag{3}$$

$\boldsymbol{h}_v^{(0)} = \boldsymbol{x}_v$ where $\boldsymbol{x}_v$ is the initial feature vector for the node. Previous works on graph neural networks for zero-shot learning have used GloVe (Pennington et al., 2014) to represent the initial features (Wang et al., 2018). Finally, Hamilton et al. (2017a) proposed using LSTMs as non-linear aggregators. However, their outputs can be sensitive to the order of the neighbours in the input, i.e., they are not permutation invariant. For example, on the Animals with Attributes 2 dataset, we find that when given the same test image 10 times with different neighbourhood orderings, an LSTM-based graph neural network outputs inconsistent predictions 16% of the time (Appendix D). One recent work considers trying to make LSTMs less sensitive by averaging the outputs over permutations, but this significantly increases the computational cost and provides only a small boost to prediction accuracy Murphy et al. (2019). In contrast, our proposed TrGCN in ZSL-KG is non-linear and naturally permutation invariant.

## 4 The ZSL-KG Framework

Here we introduce ZSL-KG: a framework with a novel transformer graph convolutional network (TrGCN) to learn class representation from common sense knowledge graphs.

Common sense knowledge graphs organize high-level knowledge implicit to humans in a graph. The nodes in the graph are concepts associated with each other via edges. These associations in the graph offer a rich and a large-scale source of high-level information, which makes them applicable to a wide range of tasks. To learn class representations, we look to existing zero-shot learning frameworks with graph neural networks. Existing zero-shot learning frameworks such as GCNZ (Wang et al., 2018) and DGP (Kampffmeyer et al., 2019) that learn class representations from structured knowledge are applicable only to small graphs like ImageNet or WordNet as they make restrictive assumptions. First, GCNZ requires the full graph Laplacian be known during training and performs several expensive computations that do not scale well. GCNZ computes the graph Fourier transform that requires multiplication of the node features with the eigenvector matrix of the graph Laplacian and computing the eigenvector matrix itself is computationally expensive (Kipf & Welling, 2017). Since publicly available common sense knowledge graphs range roughly from 100,000 to 8 million nodes and 2 million to 21 million edges (Speer et al., 2017; Zhang et al., 2020), computing the graph Fourier

transform for such graphs is impractical. Further, in zero-shot learning tasks, the graph may change at test time as new classes are added. Second, DGP requires a directed acyclic graph or parent-child relationship in the graph. In a common sense knowledge graph, we are not restricted to parent-child relationships.

To overcome these limitations, we propose to learn class representations with a novel graph neural network: transformer graph convolutional networks (TrGCN). Transformers (Vaswani et al., 2017) are non-linear modules typically used for machine translation and language modeling tasks. They achieve a non-linear combination of the input sets using multilayer perceptrons and self attention. We exploit this property to learn a permutation invariant non-linear aggregator that captures the complex structure of a common sense knowledge graph. Finally, unlike GCNZ, TrGCN is inductive and learns node representations by aggregating the local neighbourhood features. This means the learned model can be used to predict with new graph structures without retraining, which makes them well-suited for zero-shot learning.

Here we describe TrGCN (see Figure 1). We pass the neighbourhood node features $\boldsymbol{h}_u^{(l-1)}$ through a two-layer perceptron with a ReLU activation between the layers. The previous features are added to its output features with a skip connection, followed by layer normalization (LN) (Ba et al., 2016):

$$\boldsymbol{h}'^{(l-1)}_u = \text{LN}\left\{ \boldsymbol{W}_{fh}^{(l)} \cdot \left[ \sigma\left( \boldsymbol{W}_{hf}^{(l)} \cdot \boldsymbol{h}_u^{(l-1)} \right) \right] + \boldsymbol{h}_u^{(l-1)} \right\} \tag{4}$$

where $\boldsymbol{W}_{hf}^{(l)} \in \mathbb{R}^{d_{(l-1)} \times d_{(f)}}$ and $\boldsymbol{W}_{fh}^{(l)} \in \mathbb{R}^{d_{(f)} \times d_{(l-1)}}$ are learnable weight matrices for the feedforward neural network and $\sigma(.)$ is ReLU activation. The non-linear neighbourhood features are then passed through the self attention layer to compute the weighted combination of the features for each query node:

$$\left\{ \boldsymbol{z}_u^{(l)} \ \forall u \in \mathcal{N}(v) \right\} = \text{softmax}\left( \frac{\boldsymbol{Q}\boldsymbol{K}^T}{\sqrt{d_{(p)}}} \right) \boldsymbol{V} \tag{5}$$

where $\boldsymbol{Q} = \boldsymbol{W}_q^{(l)} \cdot \boldsymbol{h}'^{(l-1)}_u$ is the set of all neighbourhood query vectors, $\boldsymbol{K} = \boldsymbol{W}_k^{(l)} \cdot \boldsymbol{h}'^{(l-1)}_u$ is the set of all key vectors, $\boldsymbol{V} = \boldsymbol{W}_v^{(l)} \cdot \boldsymbol{h}'^{(l-1)}_u$ is the set of values vectors, and $\boldsymbol{W}_q \in \mathbb{R}^{d_{(l-1)} \times d_{(p)}}$, $\boldsymbol{W}_k \in \mathbb{R}^{d_{(l-1)} \times d_{(p)}}$, $\boldsymbol{W}_v \in \mathbb{R}^{d_{(l-1)} \times d_{(p)}}$ are learnable weight matrices with the projection dimension $d_{(p)}$. The output features from the attention layer is projected with another linear layer and added to its previous features with a skip connection, followed by layer normalization:

$$\left\{ \boldsymbol{z}'^{(l)}_u \ \forall u \in \mathcal{N}(v) \right\} = \text{LN}\left( \boldsymbol{W}_z^{(l)} \cdot \boldsymbol{z}_u^{(l-1)} + \boldsymbol{h}'^{(l-1)}_u \right) \tag{6}$$

where $\boldsymbol{W}_z^{(l)} \in \mathbb{R}^{d_{(p)} \times d_{(l-1)}}$ is a learnable weight matrix.

To get the aggregated vector $\boldsymbol{a}_v^{(l)}$ for node $v$, we pass the output vectors $\{ \boldsymbol{z}'^{(l)}_u \ \forall u \in \mathcal{N}(v) \}$ from the transformer encoder through a permutation invariant pooling function $\mu(.)$ such as mean-pooling. The aggregated vector is passed through a linear layer followed by a non-linearity $\sigma(.)$ such as ReLU or LeakyReLU:

$$\boldsymbol{a}_v^{(l)} = \mu\left( \left\{ \boldsymbol{z}'^{(l)}_u \ \forall u \in \mathcal{N}(v) \right\} \right) \quad \boldsymbol{h}_v^{(l)} = \sigma\left( \boldsymbol{W}^{(l)} \cdot \boldsymbol{a}_v^{(l)} \right) \tag{7}$$

where $\boldsymbol{W}^{(l)} \in \mathbb{R}^{d_{(l-1)} \times d_{(l)}}$ is a learnable weight matrix.

**Differences between TrGCN and GAT.** Joshi (2020) has drawn parallels between transformers and graph attention networks (GAT), suggesting that they are equivalent. Their work shows GAT is equivalent to transformers if the graph is treated as fully connected. While there are similarities between GAT and TrGCN, there are also key differences. GAT applies self attention to compute scalar weights for $|\mathcal{N}(v)|$ and takes the linear combination of the scalar weights and $|\mathcal{N}(v)|$ neighbours with a complexity of $O(|\mathcal{N}(v)|)$ to get the node representation. In contrast, TrGCN applies self attention to all node pairs in the neighbourhood to compute new non-linear representations and then aggregates with a pooling function to get a node representation with a computational complexity of $O(|\mathcal{N}(v)|^2)$. This design makes TrGCN more computationally expensive than other inductive graph convolutional networks such as GAT, but TrGCN can often improve accuracy across tasks (see Table 8). We also include the differences in the resource requirements for TrGCN and GAT in Appendix C Finally, in Section 6, we show that TrGCN performs better than GAT with more heads, suggesting that our self attention and non-linear aggregation contributes to the difference in performance.

**Neighbourhood Sampling.** In our experiments, we use ConceptNet (Speer et al., 2017) as our common sense knowledge graph, but ZSL-KG is agnostic to the choice of the knowledge graph. ConceptNet has high node degree, which poses a challenge to train the graph neural network. To solve this problem, we explored numerous neighbourhood sampling strategies. Existing work on sampling neighbourhood includes random sampling (Hamilton et al., 2017a), importance sampling (Chen et al., 2018a), random walks (Ying et al., 2018), etc. Similar to PinSage (Ying et al., 2018), we simulate random walks for the nodes in the graph and assign hitting probabilities to the neighbourhood nodes. During training and testing the graph neural network, we select the top $N$ nodes from the neighbourhood based on their hitting probability.

**Stacked Calibration.** In generalized zero-shot learning setting, we predict both seen and unseen classes. However, generalized ZSL models tend to overpredict the seen classes (Chao et al., 2016). Following prior work (Xu et al., 2020), we solve this issue by simply lowering the compatibility scores for the seen classes:

$$f'(\theta(x), \boldsymbol{W}, \phi(y_i) = f(\theta(x), \boldsymbol{W}, \phi(y_i)) - \gamma \mathbf{1}_{y_i \in Y_S} \tag{8}$$

where the calibration coefficient $\gamma \in \mathbb{R}$ is a hyperparameter tuned on the held-out seen classes.

## 5 Tasks and Results

We evaluate our framework on three zero-shot learning tasks: fine-grained entity typing, intent classification, and object classification.

In all our experiments, we compare ZSL-KG with other graph-based methods: GCNZ, SGCN, and DGP. GCNZ (Wang et al., 2018) uses symmetrically normalized graph Laplacian to generate the class representations. SGCN, introduced as a baseline in Kampffmeyer et al. (2019), uses an asymmetrical normalized graph Laplacian to learn the class representations. Finally, DGP (Kampffmeyer et al., 2019) uses a dense graph connectivity scheme with a two-stage propagation from ancestors and descendants to learn the class representations.

In each task, we also evaluate ZSL-KG against the specialized state-of-the-art methods. These methods are considered specialized because they either cannot easily be adapted to multiple domains (e.g., attribute-based methods cannot easily be adapted to language dataset) or require greater human effort to make them applicable to other domains (e.g., mapping ImageNet to Wikipedia articles). The code for our experiments has been released[1].

**Setup.** We use two-layer graph neural networks for the graph-based methods and ZSL-KG. We mapped the classes to the nodes in the WordNet graph for each dataset and use the code obtained from Kampffmeyer et al. (2019) to adapt the methods to language datasets. We provide details related to the mapping of classes to ConceptNet, post-processing ConceptNet, sampling, and random walk details in Appendix E and the pseudocode for ZSL-KG in Appendix F. Additional task-specific details are below.

### 5.1 Fine-Grained Entity Typing

Fine-grained entity typing is the task of categorizing the entities of a sentence into one or more narrowly scoped semantic types. The ability to identify novel fine-grained types without additional human effort would benefit several downstream tasks such as relation extraction (Yavuz et al., 2016), and coreference resolution (Durrett & Klein, 2014).

**Datasets.** Fine-grained entity typing is a zero-shot multi-label classification task because each entity can be associated with more than one type. We evaluate on popular fine-grained entity typing datasets: OntoNotes (Gillick et al., 2014) and BBN (Weischedel & Brunstein, 2005). We split the dataset into two: coarse-grained labels (e.g., `/location`) and fine-grained labels (e.g., `/location/city`). See Appendix G for more details on the datasets. Following the prior work (Obeidat et al., 2019), we train on the coarse-grained labels and predict on both coarse-grained and fine-grained labels in the test set. J.1.

**Experiment.** We use a bilinear similarity function (Eq. 1) for ZSL-KG and the other graph-based methods. The example encoder is AttentiveNER biLSTM (Shimaoka et al., 2017) (see Appendix H) and the class

---

[1]https://github.com/BatsResearch/nayak-tmlr22-code

| | *OntoNotes* | | | *BBN* | | | Overall |
|---|---|---|---|---|---|---|---|
| | Strict | Loose Mic. | Loose Mac. | Strict | Loose Mic. | Loose Mac. | Avg. Strict |
| GCNZ | $41.46 \pm 0.81$ | $54.61 \pm 0.52$ | $61.65 \pm 0.52$ | $21.47 \pm 1.22$ | $47.17 \pm 1.04$ | $56.81 \pm 0.68$ | 31.47 |
| SGCN | $42.64 \pm 0.69$ | $\mathbf{56.25 \pm 0.29}$ | $62.93 \pm 0.39$ | $24.91 \pm 0.24$ | $50.02 \pm 0.73$ | $\mathbf{59.68 \pm 0.54}$ | 33.77 |
| DGP | $41.11 \pm 0.78$ | $54.08 \pm 0.62$ | $61.38 \pm 0.76$ | $23.99 \pm 0.14$ | $47.19 \pm 0.51$ | $57.62 \pm 0.89$ | 32.55 |
| ZSL-KG | $\mathbf{45.21 \pm 0.36}$ | $55.81 \pm 0.41$ | $\mathbf{63.64 \pm 0.51}$ | $\mathbf{26.69 \pm 2.41}$ | $\mathbf{50.30 \pm 1.21}$ | $57.66 \pm 1.15$ | $\mathbf{35.95}$ |

Table 1: The results for generalized zero-shot fine-grained entity typing on Ontonotes and BBN. We report the average strict accuracy, loose micro F1, and loose macro F1 of the models on 5 random seeds and the standard error.

encoder is the graph neural network. We also reconstruct the specialized state-of-the-art methods for task: OTyper (Yuan & Downey, 2018), and DZET (Obeidat et al., 2019). OTyper uses average word embeddings and DZET uses Wikipedia Descriptors as their class representations. The methods are trained by minimizing the cross-entropy loss. For more details on the experiment, training, and inference see Appendix J.1.

As is common for this task (Obeidat et al., 2019), we evaluate the performance of our model on strict accuracy, loose micro F1, and loose macro F1 (Appendix K). Strict accuracy penalizes the model for incorrect label predictions and the number of the label predictions have to match the ground truth, whereas loose micro F1 and loose macro F1 measures if the correct label is predicted among other false positive predictions.

**Results.** Table 1 shows that, on average, ZSL-KG outperforms the best performing graph-based method (SGCN) by 2.18 strict accuracy points. SGCN has higher loose micro F1 on OntoNotes

| | *OntoNotes* | *BBN* |
|---|---|---|
| | Strict | Strict |
| OTyper | $41.72 \pm 0.44$ | $25.76 \pm 0.25$ |
| DZET | $42.88 \pm 0.47$ | $26.20 \pm 0.13$ |
| ZSL-KG | $\mathbf{45.21 \pm 0.36}$ | $\mathbf{26.69 \pm 2.41}$ |

Table 2: The results for generalized zero-shot learning on OntoNotes and BBN with specialized state-of-the-art methods.

and loose macro F1 on BBN datasets because it overpredicts labels and has greater false positives compared to the ZSL-KG. Our method has higher precision for label predictions and therefore, higher strict accuracy compared to other methods. Table 2 shows that ZSL-KG outperforms the specialized baselines and achieves the new state-of-the-art on the task.

## 5.2 Intent Classification

We next experiment on zero-shot intent classification. Intent classification is a text classification task of identifying users' intent expressed in chatbots and personal voice assistants.

**Dataset.** We evaluate on the main open-source benchmark for intent classification: SNIPS-NLU (Coucke et al., 2018). The dataset was collected using crowdsourcing to benchmark the performance of voice assistants. The training set has 5 seen classes which we split into 3 train classes and 2 development classes.

**Experiment.** Zero-shot intent classification is a multi-class classification task. The example encoder used in our experiments is a biLSTM with attention as seen in the previous section (Appendix I). We train the model for 10 epochs by minimizing the cross entropy loss and pick the model with the least loss on the development set. We measure accuracy on the test classes.

We compare ZSL-KG against existing specialized state-of-the-art methods in the literature for zero-shot intent classification: Zero-shot DNN (Kumar et al., 2017), IntentCapsNet (Xia et al., 2018), and ResCapsNet-ZS (Liu et al., 2019a). IntentCapsNet and ResCapsNet-ZS are CapsuleNet-based (Sabour et al., 2017) approaches and have reported the best performance on the task.

| | *SNIPS-NLU* |
|---|---|
| | Accuracy |
| Zero-shot DNN | 71.16 |
| IntentCapsNet | 77.52 |
| ReCapsNet-ZS | 79.96 |
| GCNZ | $82.47 \pm 03.09$ |
| SGCN | $50.27 \pm 14.13$ |
| DGP | $64.41 \pm 12.87$ |
| ZSL-KG | $\mathbf{88.98 \pm 01.22}$ |

Table 3: The results for intent classification on the SNIPS-NLU dataset.

| | | Hit@k(%) | | | | | | | | | | | | | | |
| | | 2-Hops | | | | | 3-Hops | | | | | All | | | | |
| | | 1 | 2 | 5 | 10 | 20 | 1 | 2 | 5 | 10 | 20 | 1 | 2 | 5 | 10 | 20 |
|---|---|---|---|---|---|---|---|---|---|---|---|---|---|---|---|---|
| ZSL | GCNZ | 19.8 | 33.3 | 53.2 | 65.4 | 74.6 | 4.1 | 7.5 | 14.2 | 20.2 | 27.7 | 1.8 | 3.3 | 6.3 | 9.1 | 12.7 |
| | SGCN | 26.2 | 40.4 | 60.2 | 71.9 | 81.0 | 6.0 | 10.4 | 18.9 | 27.2 | 36.9 | 2.8 | 4.9 | 9.1 | 13.5 | 19.3 |
| | DGP | **26.6** | **40.7** | 60.3 | **72.3** | **81.3** | **6.3** | 10.7 | 19.3 | 27.7 | 37.7 | **3.0** | 5.0 | 9.3 | 13.9 | 19.8 |
| | ZSL-KG | 26.3 | 40.6 | 60.3 | 71.9 | 81.2 | **6.3** | **11.1** | **20.1** | **28.8** | **38.8** | **3.0** | **5.3** | **9.9** | **14.8** | **21.0** |
| GZSL | GCNZ | 9.7 | 20.4 | 42.6 | 57.0 | 68.2 | 2.2 | 5.1 | 11.9 | 18.0 | 25.6 | 1.0 | 2.3 | 5.3 | 8.1 | 11.7 |
| | SGCN | **11.9** | **27.0** | **50.8** | 65.1 | 75.9 | 3.2 | 7.1 | 16.1 | 24.6 | 34.6 | 1.5 | 3.4 | 7.8 | 12.3 | 18.2 |
| | DGP | 10.3 | 26.4 | 50.3 | **65.2** | **76.0** | 2.9 | 7.1 | 16.1 | 24.9 | 35.1 | 1.4 | 3.4 | 7.9 | 12.6 | 18.7 |
| | ZSL-KG | 11.1 | 26.2 | 50.0 | 64.3 | 75.3 | **3.4** | **7.5** | **16.9** | **26.1** | **36.5** | **1.7** | **3.8** | **8.5** | **13.5** | **19.9** |

Table 5: The results for object classification on ImageNet dataset. We report the class-balanced top-k accuracy on zero-shot learning (ZSL) and generalized zero-shot learning (GZSL) for ImageNet classes k-hops away from the ILSVRC 2012 classes. The results for GCNZ, SGCN, and DGP are obtained from Kampffmeyer et al. (2019)

| | *AWA2* | | | | *aPY* | | | | *Overall* |
| | T1 | U | S | H | T1 | U | S | H | Avg. H |
|---|---|---|---|---|---|---|---|---|---|
| GCNZ | $77.00 \pm 2.05$ | $66.62 \pm 1.28$ | $81.63 \pm 0.18$ | $73.30 \pm 0.75$ | $51.66 \pm 0.74$ | $49.11 \pm 0.39$ | $\mathbf{71.26 \pm 0.20}$ | $58.14 \pm 0.29$ | 65.74 |
| SGCN | $77.10 \pm 1.49$ | $67.51 \pm 1.29$ | $81.16 \pm 0.11$ | $73.67 \pm 0.74$ | $52.32 \pm 0.37$ | $47.29 \pm 0.63$ | $71.05 \pm 0.18$ | $56.78 \pm 0.47$ | 65.23 |
| DGP | $77.10 \pm 1.33$ | $\mathbf{71.26 \pm 1.02}$ | $79.39 \pm 0.09$ | $\mathbf{75.09 \pm 0.61}$ | $49.73 \pm 0.30$ | $46.16 \pm 0.66$ | $70.16 \pm 0.25$ | $55.68 \pm 0.52$ | 65.38 |
| ZSL-KG | $\mathbf{78.08 \pm 0.84}$ | $66.80 \pm 0.70$ | $\mathbf{84.42 \pm 0.33}$ | $74.58 \pm 0.53$ | $\mathbf{60.54 \pm 0.58}$ | $\mathbf{55.16 \pm 0.50}$ | $69.66 \pm 0.52$ | $\mathbf{61.57 \pm 0.45}$ | **68.07** |

Table 6: The results for generalized zero-shot object classification on the AWA2 and aPY dataset. We report the average class-balanced accuracy of the models for ZSL (T1) and GZSL (U, S, H) on 5 random seeds and the standard error.

**Results.** Table 3 shows the results. ZSL-KG significantly outperforms the existing approaches and improves the state-of-the-art accuracy to 88.98%. The graph-based methods have mixed performance on intent classification and suggest that ZSL-KG works well on a broader range of tasks.

### 5.3 Object Classification

Object classification is the computer vision task of categorizing objects.

**Datasets.** Zero-shot object classification is a multiclass classification task. We evaluate our method on the large-scale ImageNet (Deng et al., 2009), Attributes 2 (AWA2) (Xian et al., 2018b), and attribute Pascal Yahoo (aPY) (Farhadi et al., 2009) datasets. See Appendix G for more details on the datasets.

**Experiment.** Following prior work (Kampffmeyer et al., 2019; Wang et al., 2018), we learn class representations by minimizing the L2 distance between the learn class representations and the weights of the fully connected layer of a ResNet classifier pretrained on the ILSVRC 2012. Next, we freeze the class representations and finetune the ResNet-backbone on the training images from the dataset. More details on the experiments are included in Appendix J.2.

| | *AWA2* | | | | *aPY* | | | |
| | T1 | U | S | H | T1 | U | S | H |
|---|---|---|---|---|---|---|---|---|
| ZSML 2020 | 77.5 | 58.9 | 74.6 | 65.8 | **64.0** | 36.3 | 46.6 | 40.9 |
| APNet 2020a | - | 54.8 | 83.9 | 66.4 | - | 32.7 | **74.7** | 45.5 |
| AGZSL 2021 | 76.4 | **69.0** | 86.5 | **76.8** | 43.7 | 36.2 | 58.6 | 44.8 |
| DPPN 2021 | - | 63.1 | **86.8** | 73.1 | - | 40.0 | 61.2 | 48.4 |
| TransZero++ 2021 | 72.6 | 64.6 | 82.7 | 72.5 | - | - | - | - |
| MSDN 2022b | 70.1 | 62.0 | 74.5 | 67.7 | - | - | - | - |
| ZSL-KG | **78.1** | 66.8 | 84.4 | 74.6 | 60.5 | **55.2** | 69.7 | **61.6** |

Table 4: The results for generalized zero-shot object classification on the AWA2 and aPY dataset with best performing specialized methods.

Following prior work (Kampffmeyer et al., 2019), we evaluate ImageNet on two settings: zero-shot learning (ZSL) where the model predicts only unseen classes, and generalized zero-shot

learning (GZSL) where the model predicts both seen and unseen classes. We follow the train/test split from Frome et al. (2013), and evaluate ZSL-KG on three levels of difficulty: 2-hops (1549 classes), 3-hops (7860 classes), and All (20842 classes). The hops refer to the distance of the classes from the ILSVRC train classes. We report the class-balanced top-K (Hit@k) accuracy for each of the hops.

We evaluate AWA2 and aPY in the ZSL and GZSL settings as well. Following prior work (Xian et al., 2018b), for ZSL, we use the pretrained ResNet101 as the backbone and report the class-balanced accuracy on the unseen classes (T1). Following prior work (Min et al., 2020), for GZSL, we use the finetuned ResNet101 to report the class-balanced accuracy on the unseen classes (U), seen classes (S), and their harmonic mean (H).

**Results.** Table 6 shows that ZSL-KG outperforms existing graph-based methods on aPY dataset by 3.53 points on the harmonic mean and shows an average improvement across both the datasets by an average of 2.33 points on the harmonic mean metric. We also observe that the graph-based methods show a significant drop in accuracy from AWA2 to aPY, whereas our method consistently achieves high accuracy on both datasets.

Table 4 shows ZSL-KG compared with specialized attribute-based methods. Our results show that we significantly outperform existing attribute-based methods on the aPY datasets and show competitive performance on the AWA2 dataset. We suspect that pretraining ZSL-KG class representations on ILSVRC 2012 classes helps the performance. Existing attribute-based methods cannot be pretrained on ILSVRC because ILSVRC does not have hand-crafted attributes. This highlights the potential benefits of using class representations from graphs. Furthermore, we note that AGZSL (Chou et al., 2021) and ZSML (Verma et al., 2020) allow the model to access the attributes and names of the unseen classes during training, which makes them transductive. Nonetheless, we either outperform them or achieve comparable performance. Finally, we include extended results with other specialized methods in Appendix A.

Table 5 shows results for ImageNet dataset. ZSL-KG outperforms existing graph-based methods on 3-hops (7860 classes) and All (20842 classes) for ZSL and GZSL evaluation. ZSL-KG despite being trained on a noisier graph, achieves competitive performance on the ImageNet dataset on both ZSL and GZSL settings.

### 5.4 Discussion

Overall, our results show that ZSL-KG improves over existing WordNet-based methods on OntoNotes, BBN, SNIPS-NLU, aPY, and ImageNet (all test classes). DGP does slightly better on AWA2, but it performs relatively poorly on aPY and ImageNet. In contrast, we see that ZSL-KG achieves the highest performance on larger test sets of ImageNet and achieves a new state-of-the-art for aPY, while maintaining competitive performance on AWA2. Finally, averaging the strict accuracy on OntoNotes and BBN, harmonic mean on AWA2 and aPY, top-1 GZSL accuracy for all classes on ImageNet, and accuracy on SNIPS-NLU, we observe that ZSL-KG outperforms the best performing WordNet-based method (GCNZ) by an average of 3.5 accuracy points and all the WordNet-based methods by an average of 2.4 accuracy points. These results demonstrate the superior flexibility and generality of ZSL-KG to achieve competitive performance on both language and vision tasks.

## 6   Comparison of Graph Aggregators

We conduct an ablation study with different aggregators with our framework. Existing graph neural networks include GCN (Kipf & Welling, 2017), GAT (Veličković et al., 2018), RGCN (Schlichtkrull et al., 2018), and LSTM (Hamilton et al., 2017a). We provide all the architectural details in Appendix L. We train these models with the same experimental setting for the tasks mentioned in their respective sections.

**ZSL-KG with different aggregators.** Table 7 shows results for our ablation study. Our results show that TrGCN often outperforms existing graph neural networks with linear aggregators and TrGCN adds up to 1.23 accuracy points improvement on these tasks. We observe that GAT and GCN outperform TrGCN on BBN and ImageNet datasets. However, no architecture is superior on all tasks, but TrGCN is the best on a majority of them. With relational aggregators (ZSL-KG-RGCN), we observe that they do not outperform

| | *OntoNotes* | *BBN* | *SNIPS-NLU* | *AWA2* | *aPY* | ImageNet |
|---|---|---|---|---|---|---|
| | Strict | Strict | Acc. | H | H | All (%) |
| ZSL-KG | **45.21** | 26.69 | **88.98** | **74.58** | **61.57** | 1.74 |
| -GCN | 42.19 | 27.44 | 84.78 | 73.08 | 60.45 | **1.78** |
| -GAT | 43.48 | **32.65** | 87.57 | 73.35 | 60.91 | 1.77 |
| -RGCN | 43.88 | 27.89 | 87.47 | 47.96 | 31.60 | — |
| -LSTM | 44.52 | 26.77 | 88.81 | 55.55 | 57.02 | 0.83 |

Table 7: The results for zero-shot learning with alternate graph neural networks as class encoders in ZSL-KG.

| | *AWA2* | *aPY* |
|---|---|---|
| | H | H |
| ZSL-KG-GAT (1-head) | 73.35 | 60.91 |
| ZSL-KG-GAT (2-heads) | 72.68 | 60.72 |
| ZSL-KG-GAT (3-heads) | 71.71 | 60.65 |
| ZSL-KG (TrGCN) | **74.58** | **61.57** |

Table 9: The results for generalized zero-shot learning on AWA2 and aPY datasets with multiple heads in graph attention networks.

ZSL-KG and may reduce the overall performance (as seen in AWA2 and aPY). ZSL-KG-LSTM which uses an LSTM-based aggregator shows inconsistent performance across different tasks.

We also compare the ZSL-KG with TrGCN and other graph aggregators on conventional zero-shot learning, where only unseen classes are present during testing. Table 8 shows that ZSL-KG with TrGCN outperforms other graph neural networks on two out of the three object classification datasets.

**Comparison of WordNet and ConceptNet.** It is also worth understanding the effect of the knowledge graphs used for zero-shot learning. We compare SGCN and ZSL-KG-GCN, as they use the same linear aggregator to learn the class representation but train with different knowledge graphs, i.e. SGCN uses WordNet whereas ZSL-KG-GCN uses ConceptNet. We see that ZSL-KG-GCN trained on common sense knowledge graphs adds an improvement as high as 6.7 accuracy points across the tasks suggesting that the choice of knowledge graphs is crucial for downstream performance.

| | *AWA2* | *aPY* | *ImageNet* |
|---|---|---|---|
| | T1 | T1 | All (%) |
| ZSL-KG | **78.08** | **60.54** | 3.01 |
| -GCN | 74.81 | 57.76 | 2.97 |
| -GAT | 75.29 | 59.06 | **3.08** |
| -RGCN | 66.27 | 29.51 | — |
| -LSTM | 66.46 | 50.84 | 2.65 |

Table 8: The results for zero-shot learning tasks with other graph neural networks.

We further investigate the differences between WordNet and ConceptNet by training TrGCN with WordNet (see Appendix B). We show that TrGCN can benefit existing applications with WordNet, but might tend to work better with a richer graph structure such as ConceptNet.

**Comparison of TrGCN and GAT with multiple heads.** To better understand the differences between GAT and TrGCN, we perform an ablation with multihead attention in GAT to increase the expressivity in the ZSL-KG framework. The multihead attention in GAT is similar to Vaswani et al. (2017) but instead average the output from the heads to keep the same output dimension for the node representations. Table 9 shows that adding more heads to ZSL-KG-GAT hurts performance, while ZSL-KG with TrGCN achieves the highest performance. This suggests that the self attention and non-linear aggregator in TrGCN contributes to the difference in performance.

## 7 Conclusion

ZSL-KG is a flexible framework for zero-shot learning with common sense knowledge graphs and can be adapted to a wide variety of tasks without requiring additional annotation effort. Our framework introduces a novel transformer graph convolutional network to learn rich representations from common sense knowledge graphs. Our work demonstrates that common sense knowledge graphs are a source of high-level knowledge that can benefit many tasks.

## Acknowledgements

We thank Yang Zhang for help preparing the Ontonotes dataset. We thank Roma Patel, Elaheh Raisi, Charles Lovering, and our anonymous reviewers for providing helpful feedback on our work. This material is based on research sponsored by Defense Advanced Research Projects Agency (DARPA) and Air Force Research Laboratory (AFRL) under agreement number FA8750-19-2-1006. The U.S. Government is authorized to reproduce and distribute reprints for Governmental purposes notwithstanding any copyright notation thereon. The views and conclusions contained herein are those of the authors and should not be interpreted as necessarily representing the official policies or endorsements, either expressed or implied, of Defense Advanced Research Projects Agency (DARPA) and Air Force Research Laboratory (AFRL) or the U.S. Government. We gratefully acknowledge support from Google and Cisco. Disclosure: Stephen Bach is an advisor to Snorkel AI, a company that provides software and services for weakly supervised machine learning.

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

| | AWA2 | | | | aPY | | | |
|---|---|---|---|---|---|---|---|---|
| | T1 | U | S | H | T1 | U | S | H |
| SP-AEN (Chen et al., 2018b) | 58.5 | 23.3 | 90.9 | 37.1 | 24.1 | 13.7 | 63.4 | 22.6 |
| PSR (Annadani & Biswas, 2018) | 63.8 | 20.7 | 73.8 | 32.3 | 38.4 | 13.5 | 51.4 | 21.4 |
| Relation Net (Sung et al., 2018) | 64.2 | 30.0 | 93.4 | 45.3 | - | - | - | - |
| LFGAA+Hibrid (Liu et al., 2019b) | 68.1 | 27.0 | 93.4 | 41.9 | - | - | - | - |
| AREN (Xie et al., 2019) | 66.9 | 54.7 | 79.1 | 64.7 | 39.2 | 30.0 | 47.9 | 36.9 |
| f-VAEGAN-D2 (Xie et al., 2019) | 70.3 | 57.1 | 76.1 | 65.2 | - | - | - | - |
| ZSML (Verma et al., 2020) | 77.5 | 58.9 | 74.6 | 65.8 | **64.0** | 36.3 | 46.6 | 40.9 |
| GXE (Li et al., 2019) | 71.1 | 56.4 | 81.4 | 66.7 | 38.0 | 26.5 | 74.0 | 39.0 |
| OCD-CVAE (Keshari et al., 2020) | 71.3 | 59.5 | 73.4 | 65.7 | - | - | - | - |
| E-PGN (Yu et al., 2020) | 73.4 | 52.6 | 83.5 | 64.6 | - | - | - | - |
| LsrGAN (Vyas et al., 2020) | - | 54.6 | 74.6 | 63.0 | - | - | - | - |
| TF-VAEGAN (Narayan et al., 2020) | 73.4 | 55.5 | 83.6 | 66.7 | - | - | - | - |
| APN (Xu et al., 2020) | 71.7 | 62.2 | 69.5 | 65.6 | - | - | - | - |
| APNet (Liu et al., 2020a) | - | 54.8 | 83.9 | 66.4 | - | 32.7 | **74.7** | 45.5 |
| DBVE (Min et al., 2020) | - | 62.7 | 77.5 | 69.4 | - | 37.9 | 55.9 | 45.2 |
| CN-ZSL (Skorokhodov & Elhoseiny, 2021) | - | 60.2 | 77.1 | 67.6 | - | - | - | - |
| CE-GZSL (Han et al., 2021) | 70.4 | 63.1 | 78.6 | 70.0 | - | - | - | - |
| IPN (Liu et al., 2021a) | - | 67.5 | 79.2 | 72.9 | - | 37.2 | 66 | 47.6 |
| AGZSL (Chou et al., 2021) | 76.4 | 69.0 | 86.5 | **76.8** | 43.7 | 36.2 | 58.6 | 44.8 |
| DPPN (Wang et al., 2021) | - | 63.1 | **86.8** | 73.1 | - | 40.0 | 61.2 | 48.4 |
| RSR (Liu et al., 2021b) | 68.4 | 55.3 | 76.0 | 64.0 | 45.4 | 31.3 | 50.9 | 38.7 |
| TransZero (Chen et al., 2022a) | 70.1 | 61.3 | 82.3 | 70.2 | - | - | - | - |
| TransZero++ (Chen et al., 2021) | 72.6 | 64.6 | 82.7 | 72.5 | - | - | - | - |
| ERPCNet (Li et al., 2022) | 71.8 | 59.1 | 82.0 | 68.7 | 43.5 | 32.7 | 49.3 | 39.3 |
| MSDN (Chen et al., 2022b) | 70.1 | 62.0 | 74.5 | 67.7 | - | - | - | - |
| GCNZ (Wang et al., 2018) | 77.0 | 66.6 | 81.6 | 73.3 | 51.7 | 49.1 | 71.3 | 58.1 |
| SGCN Kampffmeyer et al. (2019) | 77.1 | 67.5 | 81.2 | 73.7 | 52.3 | 47.3 | 71.1 | 56.8 |
| DGP Kampffmeyer et al. (2019) | 77.1 | **71.3** | 79.4 | 75.1 | 49.7 | 46.2 | 70.2 | 55.7 |
| ZSL-KG (ours) | **78.1** | 66.8 | 84.4 | 74.6 | 60.5 | **55.2** | 69.7 | **61.6** |

Table 10: ZSL-KG compared to attribute-based zero-shot learning methods.

## A    Results on AWA2 and aPY

Table 10 shows results comparing ZSL-KG with other related work from zero-shot object classification.

## B    TrGCN with WordNet

To further illustrate the importance of using a richer knowledge graph such as ConceptNet, we experiment with TrGCN trained on the WordNet graph. Table 11 shows that TrGCN with WordNet on OntoNotes outperforms other WordNet-based methods but underperforms the best-performing WordNet-based method on the rest of the datasets. However, we see that ZSL-KG, i.e., TrGCN with ConceptNet always improves the performance compared to TrGCN with WordNet. These inconsistent results suggest that TrGCN can benefit existing applications with WordNet, but might tend to work better with a richer graph structure such as ConceptNet.

|  | *OntoNotes* | *BBN* | *SNIPS-NLU* | *AWA2* | *aPY* |
|---|---|---|---|---|---|
|  | Strict | Strict | Acc. | H | H |
| GCNZ | 41.50 | 21.50 | 82.47 | 73.30 | 58.10 |
| SGCN | 42.60 | 24.90 | 50.30 | 73.70 | 56.80 |
| DGP | 41.11 | 23.99 | 64.41 | **75.10** | 55.70 |
| TrGCN (WordNet) | 44.42 | 23.09 | 41.85 | 72.16 | 55.39 |
| ZSL-KG (ConceptNet) | **45.21** | **26.69** | **88.98** | 74.58 | **61.57** |

Table 11: The results showing zero-shot performance of existing WordNet-based methods, TrGCN with WordNet, and ZSL-KG, i.e., TrGCN with ConceptNet.

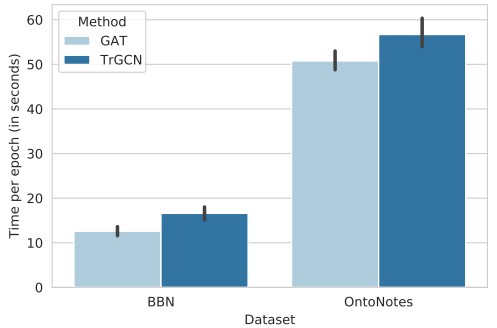
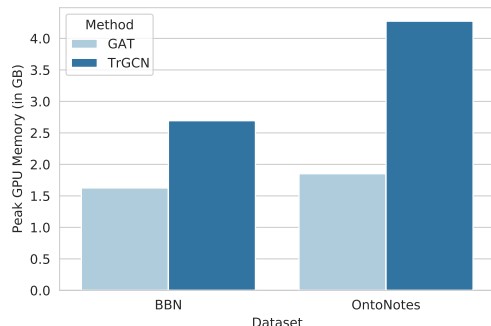

Figure 2: Graphs showing time taken per epoch (in seconds) and peak GPU memory (in GB) for fine-grained entity typing datasets.

## C  Resource Requirements for ZSL-KG with TrGCN and GAT

Here we measure differences between TrGCN and GAT in terms of resource requirement. In particular, we perform experiments to compute the average training time per epoch and GPU memory requirements. We run experiments with the fine-grained entity typing datasets, namely BBN and OntoNotes. We use the same hyperparameters as mentioned in Appendix M and run our experiments on an NVIDIA RTX 3090 with 24GB of GPU memory.

Our results in Figures 2 show that, during training, TrGCN takes slightly longer time per epoch and greater GPU memory compared to GAT. We note that both GAT and TrGCN benefit from the importance sampling as they can be batched and processed efficiently. On the other hand, GCNZ results in out-of-memory (OOM) error as it requires the full graph during training and testing.

## D  LSTM Predictions

LSTMs have been used in graph neural networks as aggregators to generate more expressive node embeddings. However, LSTMs assume an ordering of the inputs which is not present in a graph neighbourhood. To apply LSTMs to an unordered set of nodes, Hamilton et al. (2017a) randomly permute the nodes in the neighbourhood.

We test whether randomly permuting the node neighbours makes LSTM-based aggregator permutation invariant. We replicate the LSTM-based aggregator to work with the ZSL-KG framework. The model is trained with the setup described in Section 5.3. We run the prediction on the Animals with Attributes 2 dataset by computing 10 class representation for each of the classes using the trained LSTM-based aggregator model.

The experiments reveal that 1325 out of 7913 (16.78%) have multiple predictions for the same image in the unseen classes. For images that have multiple predictions, we take the count of the mode prediction and plot

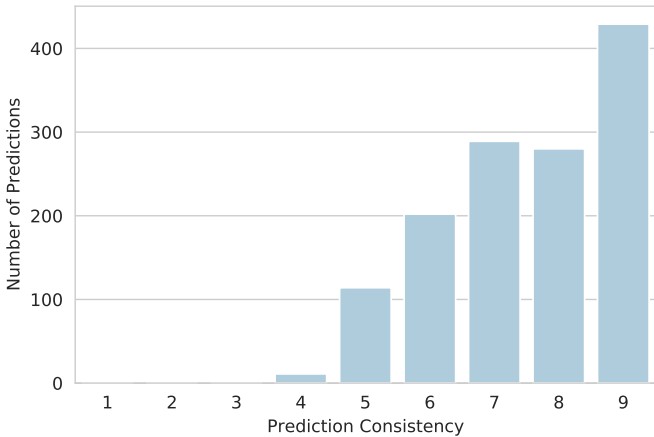

Figure 3: Graph showing distribution of inconsistencies in the LSTM-based aggreagtor predictions.

the histogram. Figure 3 shows inconsistency in predictions. The graph for a given value $p$ on the x-axis is read as for every 10 prediction, $p$ times the same output is predicted.

## E ConceptNet Setup

In all our experiments with ZSL-KG, we map each class to a node in ConceptNet 5.7 (Speer et al., 2017) and query its 2-hop neighbourhood. Next, we remove all the non-English concepts and their edges from the graph and make all the edges bidirectional. Then, we optionally take the union of the concepts' neighbourhood that share the same prefix noun prefix. For example, we take the union of the neighbourhood nodes of `/c/en/lawyer` and `/c/en/lawyer/n`. Then, we compute the embeddings for the concept using the pretrained 300 dimensional GloVe 840B (Pennington et al., 2014). We average the individual word in the concept to get the embedding. These embeddings serve as initial features for the graph neural network.

For the random walk, the number of steps is 20, and the number of restarts is 10. We add one smoothing to the visit counts and normalize the counts for the neighboring nodes. During training, we sample K neighbours with the highest hitting probabilities at each hop. For OntoNotes, BBN, AWA2, aPY, and SNIPS-NLU, we sample 50 neighbours in the first hop and 100 neighbours in the second hop. For ImageNet we sample 100 and 200 neighbours in the first and second hop.

## F Pseudocode

In Algorithm 1, we describe the forward pass with the ZSL-KG framework. TrGCN computes the class representations for the nodes in the graph. The class representations are used in the bilinear similarity function to compute the compatibility scores for the classes.

## G Dataset Details

Table 12 and Table 13 show the statistics for zero-shot datasets used in our experiments. Apart from these datasets, we also evaluate ZSL-KG on the ImageNet dataset. ImageNet dataset has a total of 1000 seen classes and 20842 unseen classes

We obtain the OntoNotes and BBN dataset from Ren et al. (2016). OntoNotes has three levels of types such as `/location`, `/location/structure`, `/location/structure/government` where `/location` and `/location/structure` are treated as coarse-grained entity types and `/location/-structure/government` is treated as fine-grained entity type. Similarly, BBN has two levels of types and we consider the level two

---

**Algorithm 1:** Forward pass with the ZSL-KG framework.

---

**Input** : example $x$, example encoder $\theta(x)$, linear layer $\boldsymbol{W}$, class encoder $\phi(y)$, graph $\mathcal{G}(V, E, R)$, class nodes $\{v_y^1, v_y^2, ..., v_y^n\}$, node initialization $\boldsymbol{H} = [h^0, ..., h^v]$, depth $L$, graph neural network weights $\{\boldsymbol{W}^1, ..., \boldsymbol{W}^L\}$, transformers $\{T^1, ..., T^L\}$, neighbourhood sample sizes $\{s_1, , ..., s_L\}$,

**Output** : logits for classes $y = \{y_1, y_2, ..., y_n\}$

TrGCN($\mathbb{V}$, $l = 0$);

**if** $l = L$ **then**

   |    return $\boldsymbol{H}_\mathbb{V} \leftarrow \boldsymbol{H}(\mathbb{V})$;

**else**

   |    $i \leftarrow 0$;

   |    **for** $v \in \mathbb{V}$ **do**

   |     |    $\mathbb{N} \leftarrow \mathcal{N}(v, s_{l+1})$;

   |     |    $\boldsymbol{H}_\mathbb{N} \leftarrow$ TrGCN($\mathbb{N}$, $l + 1$);

   |     |    $\boldsymbol{Z}_\mathbb{N} \leftarrow T^{(l+1)}(\boldsymbol{H}_\mathbb{N})$;

   |     |    $\boldsymbol{a}_v^{(l+1)} \leftarrow \mathrm{mean}(\boldsymbol{Z}_\mathbb{N})$;

   |     |    $\boldsymbol{h}_v^{(l+1)} \leftarrow \sigma(\boldsymbol{W}^{(l+1)} \cdot \boldsymbol{a}_v)$;

   |     |    $\boldsymbol{H}_i^{(l+1)} = \boldsymbol{h}_v^{(l+1)}/\|\boldsymbol{h}_v^{(l+1)}\|_2$;

   |     |    $i \leftarrow i + 1$

   |    **end**

   |    return $\boldsymbol{H}^{(l+1)}$

**end**

$\phi(y) \leftarrow$ TrGCN($\{v_y^1, v_y^2, ..., v_y^n\}$);

return $\theta(x)^\mathrm{T} \boldsymbol{W} \phi(y)$;

---

| Dataset | Seen classes | Unseen classes | Seen examples | Test examples |
|---|---|---|---|---|
| OntoNotes | 40 | 32 | 220398 | 9604 |
| BBN | 15 | 24 | 85545 | 12349 |
| SNIPS-NLU | 5 | 2 | 9888 | 3914 |

Table 12: Zero-shot fine-grained entity typing (OntoNotes and BBN) and intent classification (SNIPS-NLU) datasets used in our experiments.

types as fine-grained types. Furthermore, we process the datasets by removing all the fine-grained entity types from the train set. We also remove all the examples from the train set where coarse-grained entity types are not present in the test set. We note that the OntoNotes dataset has /other type which cannot be mapped to a meaningful concept or a wikipedia article. Since /other is a coarse-grained entity type, we train a weight vector for the type and treat as class representation.

For object classification datasets, our work follows the proposed splits suggested in Xian et al. (2018b). Since aPY has multiple objects in an image, we crop objects with the bounding box information provided and use the cropped images for training and testing.

## H  AttentiveNER

Here, we describe AttentiveNER (Shimaoka et al., 2017) used as the example encoder in the fine-grained entity typing task. Each mention $m$ comprises of $n$ tokens mapped to a pretrained word embedding from GloVe 840B. We average the embeddings to obtain a single vector $\boldsymbol{v}_m$:

$$\boldsymbol{v}_m = \frac{1}{n} \sum_{j=1}^{n} \boldsymbol{m}_j \tag{9}$$

| Dataset | Seen classes | Unseen classes | Seen examples | Seen test examples | Unseen test examples |
|---------|--------------|----------------|---------------|--------------------|--------------------|
| AWA2 | 40 | 10 | 23527 | 5882 | 7913 |
| aPY | 20 | 12 | 5932 | 1483 | 7924 |

Table 13: Zero-shot object classification datasets used in our experiments.

where $\boldsymbol{m}_j \in \mathbb{R}^d$ is the pretrained embedding.

We learn the context of the mention using two biLSTM with attention layers. The left context $l$ is represented by $\{l_1, l_2, ..., l_s\}$ and the right context $r$ by $\{r_1, r_2, ..., r_s\}$ where $l_i \in \mathbb{R}^d$ and $r_j \in \mathbb{R}^d$ are the pretrained word embeddings for the left and the right context. We consider a context window size of $s$. We pass $l$ and $r$ through their separate biLSTM layers to get the hidden states $\overleftarrow{\boldsymbol{h}_l}, \overrightarrow{\boldsymbol{h}_l}$ for the left context and $\overleftarrow{\boldsymbol{h}_r}, \overrightarrow{\boldsymbol{h}_r}$ for the right context. The hidden states are passed through the attention layer to compute the attention scores. The attention layer is a two-layer feedforward neural network and computes the normalized attention for each of the hidden states $v_c \in \mathbb{R}^h$:

$$\alpha_i^l = \boldsymbol{W}_\alpha(\tanh(\boldsymbol{W}_e \begin{bmatrix} \overleftarrow{\boldsymbol{h}_i^l} \\ \overrightarrow{\boldsymbol{h}_i^l} \end{bmatrix})) \tag{10}$$

$$a_i^l = \frac{\exp\left(\alpha_i^l\right)}{\sum_i \exp\left(\alpha_i^l\right) + \sum_j \exp\left(\alpha_j^r\right)} \tag{11}$$

The scalar values are multiplied with their respective hidden states to get the final context vector representation $v_c$:

$$\boldsymbol{v}_c = \sum_{i=0}^{s} a_i^l \boldsymbol{h}_i^l + \sum_{j=0}^{s} a_j^r \boldsymbol{h}_j^r \tag{12}$$

Finally, we concatenate the context vector $\boldsymbol{v}_c$ and $\boldsymbol{v}_m$ to get the example representation $\boldsymbol{x}$.

The AttentiveNER can be replaced with a pretrained language model (Onoe et al., 2021) as the example encoder and potentially improve the performance, which we leave for future work.

## I   BiLSTM with Attention

The biLSTM with attention is used to learn class representations in DZET (Obeidat et al., 2019). The input tokens $\boldsymbol{w} = w_0, w_1, ..., w_n$ represented by pretrained embeddings are passed to the biLSTM to get the hidden states $\overrightarrow{\boldsymbol{h}}$ and $\overleftarrow{\boldsymbol{h}}$. The hidden states are concatenated to get $\boldsymbol{h} = \boldsymbol{h}_0, \boldsymbol{h}_1, ..., \boldsymbol{h}_n$. Next, they are passed to the attention module to get the scalar attention values:

$$\alpha_i = \boldsymbol{W}_\alpha \left(\tanh(\boldsymbol{W}_e \cdot \boldsymbol{h}_i)\right) \tag{13}$$

The scalar attention values are then normalized to with a softmax layer:

$$a_i = \frac{\exp\left(\alpha_i\right)}{\sum_i \exp\left(\alpha_i\right)} \tag{14}$$

The normalized scalar attention values are multiplied with their respective hidden vectors to get the final representation $\boldsymbol{t}_w$:

$$\boldsymbol{t}_w = \sum_{i=0}^{n} a_i \boldsymbol{h}_i \tag{15}$$

## J   Experiment Details

### J.1   Fine-grained Entity Typing

For all the experiments, we initialize the tokens in the example encoder with 300 dimensional GloVe 840B embeddings. We reconstruct OTyper (Yuan & Downey, 2018) and DZET (Obeidat et al., 2019) for fine-grained entity typing. Otyper averages the 300 dimensional GloVe 840B embeddings for the tokens in the entity type or the class. For DZET, we manually mapped the classes to Wikipedia articles. We pass each article's first paragraph through a learnable biLSTM with attention to learn the class representations (Appendix I).

For both OntoNotes and BBN, the methods are trained for 5 epochs by minimizing the cross-entropy loss using Adam with a learning rate 0.001. During inference, we pass the scores from the bilinear similarity model through a sigmoid and pick the labels that have a probability of 0.5 or greater as our prediction. Since we have coarse-grained types along with fine grained types, we set calibration coefficient $\gamma$ to 0.

### J.2   Object Classification

For AWA2 and aPY, we follow the same L2 training scheme and train for 1000 epochs on 950 random classes from 1000 ILSVRC 2012 classes, while the remaining 50 classes are used for validation. The model with the least loss on the validation classes is used to generate the seen and unseen class representations with the graph. Since only a subset of the seen classes for AWA2 (22 out of 40) and aPY (2 out of 20) are part of ILSVRC 2012 classes, we freeze the class representations for the seen classes and fine-tune a pretrained ResNet101-backbone on the individual datasets for 25 epochs using SGD with a learning rate 0.0001 and momentum of 0.9. We calibrate $\gamma$ on the validation splits as suggested in Xian et al. (2018b) and set $\gamma = 3.0$ for AWA2 and $\gamma = 2.0$ for aPY for all the methods.

For the ImageNet experiment, we train ZSL-KG by minimizing the L2 distance between the learned class representations and the weights fully connected layer of a ResNet50 classifier for 3000 epochs on 1000 classes from the ILSVRC 2012. Similar to DGP and SGCN, we freeze the class representations for the class representations and fine-tune the ResNet-backbone on the ILSVRC 2012 dataset for 20 epochs using SGD with a learning rate 0.0001 and momentum of 0.9. During inference, we switch the knowledge graph to the ImageNet graph and generate class representations from them. For fair comparison with other graph-based methods on ImageNet, we use ResNet50 model (He et al., 2016) in Torchvision (Marcel & Rodriguez, 2010) pretrained on ILSVRC 2012 (Russakovsky et al., 2015) and do not calibrate the outputs from ZSL-KG.

## K   Fine-grained entity typing evaluation

We follow the standard evaluation metric introduced in Ling & Weld (2012): Strict Accuracy, Loose Micro F1 and Loose Macro F1.

We denote the set of ground truth types as T and the set of predicted types as P. We use the F1 computed from the precision $p$ and recall $r$ for the evaluation metrics mentioned below:

**Strict Accuracy.** The prediction is considered correct if and only if $t_e = \hat{t_e}$:

$$p = \frac{\sum_{e \in P \cap T} \mathbf{1}(t_e = \hat{t_e})}{|P|} \tag{16}$$

$$r = \frac{\sum_{e \in P \cap T} \mathbf{1}(t_e = \hat{t_e})}{|T|} \tag{17}$$

**Loose Micro.** The precision and recall scores are computed as:

$$p = \frac{\sum_{e \in P} |t_e \cap \hat{t_e}|}{\sum_{e \in P} |\hat{t_e}|} \tag{18}$$

| Method | Aggregate | Combine |
|---|---|---|
| ZSL-KG-GCN | $\boldsymbol{a}_v^{(l)} = \text{Mean}\left(\left\{\boldsymbol{h}_u^{(l-1)}, u \in \mathcal{N}(v)\right\}\right)$ | $\boldsymbol{h}_v^{(l)} = \sigma\left(\boldsymbol{W}^{(l)}\boldsymbol{a}_v^{(l)}\right)$ |
| ZSL-KG-GAT | $\alpha_u^{(l)} = \text{Attn}\left(\left\{(\boldsymbol{h}_u'^{(l-1)}||\boldsymbol{h}'^{(l-1)})_v, u \in \mathcal{N}(v)\right\}\right)$ | $\boldsymbol{h}_v^{(l)} = \sigma(\sum_{u=1}^{\mathcal{N}(v)+1}\alpha_u^{(l)}\boldsymbol{h}_u'^{(l-1)})$ |
| ZSL-KG-RGCN | $\boldsymbol{a}_v^{(l)} = \sum_{r \in R}\sum_{j \in N(v)^r}\frac{1}{c_{i,r}}\sum_{b \in B}\alpha_{b,r}^{(l)}\boldsymbol{V}_b^{(l)}\boldsymbol{h}_j^{(l-1)}$ | $\boldsymbol{h}_v = \sigma(\boldsymbol{a}_v + \boldsymbol{W}_s^{(l)}\boldsymbol{h}_v^{(l-1)})$ |
| ZSL-KG-LSTM | $\boldsymbol{a}_v^{(l)} = \text{LSTM}^{(l)}\left(\boldsymbol{h}_u^{(l-1)}\forall u \in \mathcal{N}(v)\right)$ | $\boldsymbol{h}_v^{(l)} = \sigma\left(\boldsymbol{W}\cdot[\boldsymbol{h}_v^{(l-1)}||\boldsymbol{a}_v^{(l)}]\right)$ |

Table 14: Graph Aggregators

$$p = \frac{\sum_{e\in T}|t_e \cap \hat{t_e}|}{\sum_{e\in T}|\hat{t_e}|} \tag{19}$$

**Loose Macro.** The precision and recall scores are computed as:

$$p = \frac{1}{|P|}\sum_{e\in P}\frac{|t_e \cap \hat{t_e}|}{|\hat{t_e}|} \tag{20}$$

$$r = \frac{1}{|T|}\sum_{e\in T}\frac{|t_e \cap \hat{t_e}|}{|\hat{t_e}|} \tag{21}$$

## L   Graph Neural Networks Architecture Details

| Task | Inp. dim. | Hidden dim. | Attn. dim. $\alpha$ | Low-rank dim. $h$ |
|---|---|---|---|---|
| Intent classification | 300 | 32 | 20 | 16 |
| Fine-grained entity typing | 300 | 100 | 100 | 20 |

Table 15: Hyperparameters for the biLSTM with attention example encoder in the language related tasks

Table 14 describes the aggregator and combine function for the ablation experiments. ZSL-KG-GCN uses a mean aggregator to learn the neighbourhood structure. ZSL-KG-GAT projects the neighbourhood nodes to a new features $\boldsymbol{h}_u'^{(l-1)} = \boldsymbol{W}\boldsymbol{h}_u^{(l-1)}$. The neighbourhood node features are concatenated with self feature and passed through a self-attention module for get the attention coefficients. The attention coefficients are multiplied with the neighbourhood features to the get the node embedding for the $l$-th layer in the combine function. ZSL-KG-RGCN uses a relational aggregator to learn the structure of the neighbourhood. To avoid overparameterization from the relational weights, we perform basis decomposition of the weight vector into $B$ bases. We learn $|B|$ relational coefficients and $|B|$ weight vectors in the aggregate function and add with the self feature in combine function. ZSL-KG-LSTM uses LSTM as an aggregator to combine the neighbourhood features. The nodes in the graph are passed through an LSTM and the last hidden state is taken as the aggregated vector. The aggregated vector is concatenated with the node's previous layer feature and passed to the combine function to get the node representation.

## M   Hyperparameters

In this section, we detail the hyperparameters used in our experiments.

### M.1    Training

Our framework is built using PyTorch and AllenNLP (Gardner et al., 2018). In all our experiments, we use Adam (Kingma & Ba, 2015) to train our parameters with a learning rate of 0.001, unless provided in the experiments. We set the weight decay to $5e-04$ for OntoNotes, ImageNet, AWA2, and aPY and 0.0 for BBN. For intent classification, we experiment with a weight decay of $1e-05$ and $5e-05$. We found that weight decay of 5e-05 gives the best performance overall for all the baseline graph aggregators and 1e-05 for ZSL-KG. We set the weight decay to 0.0 when fine-tuning the ResNet backbone with SGD for ImageNet, AWA2, and aPY.

For fine-grained entity typing, we assume a low-rank for the compatibility matrix $\boldsymbol{W}$. The matrix $\boldsymbol{W} \in \mathbb{R}^{d_e \times d_c}$ is factorized into $\boldsymbol{A} \in \mathbb{R}^{h \times d_e}$ and $\boldsymbol{B} \in \mathbb{R}^{d_c \times h}$ where $d_e$ is the size of example representation, $d_c$ is the size of the the class representation, and $h$ is the size of the joint semantic space. Table 15 summarizes the hyperparameters used in the example encoders which is a biLSTM with attention or a task-specific variant of it.

### M.2    Graph Neural Networks

| Task | layer-1 | layer-2 |
|------|---------|---------|
| Object classification | 2048 | 2049 |
| Intent classification | 64 | 64 |
| Fine-grained entity typing | 128 | 128 |

Table 16: Output dimensions for the layers in the graph neural networks.

Table 16 details the output dimensions of the graph neural network layers. ZSL-KG-GAT uses LeakyReLU activation in the attention. LeakyReLU has a negative slope of 0.2. ZSL-KG-RGCN learns $B$ bases weight vectors in the baseline. We found that $B = 1$ performs the best for fine-grained entity typing and object classification. In fine-grained entity typing, the activation function after the graph neural network layer is ReLU and following prior work in object classification, the activation function is LeakyReLU with a negative slope of 0.2.

| | layer 1 | | layer 2 | |
|------|---------|---------|---------|---------|
| | $d_{(f)}$ | $d_{(p)}$ | $d_{(f)}$ | $d_{(p)}$ |
| OntoNotes | 150 | 150 | 32 | 64 |
| BBN | 250 | 150 | 32 | 64 |
| SNIPS | 150 | 150 | 32 | 32 |
| AWA2/aPY | 100 | 150 | 1024 | 1024 |
| ImageNet | 150 | 150 | 1024 | 1024 |

Table 17: The hyperparameters used in our transformer graph convolutional network.

In our transformer module, there are four hyperparameters - input dimension $d_{(l-1)}$, output dimension $d_{(l-1)}$, feedforward layer hidden dimension $d_{(f)}$, and projection dimension $d_{(p)}$. The input dimension and output dimensions are the same in the aggregator. Table 17 details the hyperparameters used in our transformer graph convolutional networks. For fine-grained entity typing, we manually tuned the hyperparameters. We tuned the hyperparameters on the held-out validation classes for object classification.

