# OpenReview forum: "Zero-Shot Learning with Common Sense Knowledge Graphs"
_TMLR — Accepted by TMLR_

### Review · Reviewer_MySv · 2022-05-20

**Summary Of Contributions:**

This manuscript proposes a new method for attribute-free zero-shot learning for NLP and vision problems by leveraging knowledge from concept graphs. They introduce a Graph Attention Transformer to model relations between nodes in a graph, where the node embedding can then be used as a class embedding (class representations). Additionally, TrGCN works with dynamic graphs and hence can be used to predict over arbitrary label spaces at test-time with no knowledge of zero-shot (unseen) classes apriori.

Compared to established notion of using attribute-based learning and reasoning over seen and unseen classes, this paper takes a new approach to directly infer the class representations from concept graphs which already have rich hierarchical relationships modeled between words.






**Broader Impact Concerns:**

None.

**Requested Changes:**

Overall, I think this paper is okay as is and doesn't require significant changes. Addition of experiments as requested above can be helpful to provide further insights into the method. Few spelling mistakes, which need to be corrected in the final version.

**Strengths And Weaknesses:**

Strengths
-------------------

1) TrGCN is a transductive and attribute-free approach, which makes it **easy to apply it to any problem** without making too many prior assumptions.
2) The modeling is general enough to be applied to a wider variety of problems in vision or NLP.
3) The paper is **well written and easy to follow**.
4) SOTA performance on few tasks on zero shot learning.
5) The overall method is **simple and has lot of potential** to unlock future developments in this space.


Clarifications requested
----------------------------

1) It is not clearly explained how or why the class embedding learned by different nodes in the graph would be different from GLoVE/Word2Vec embedding which also model the relationships between words from natural text corpora. A comparison by just using glove (that is not training the graph neural network and just using the initial embedding as is) will make this more clear.

2) The performance is almost on par with using GAT on most datasets. Additionally, GAT has time complexity O(N.$m$) for N nodes and m neighbors, while current method is O(N.$m^2$), so a more detailed analysis on why current method is better than a more simpler GAT is missing. While Table 9 indicates multiple heads might hurt accuracy for GAT, a more accurate comparison should be made using multiple heads for current methods as well.

3) If two nodes (words) have same neighbors, then according to your formulation, they have exactly the same representation. Firstly, is this possible? If so, is it a good or bad thing for the current problem?

4) Sec 5.3 - "_Table 5 shows that ZSL-KG, on average, achieves a significantly higher harmonic mean than the graph-based methods on AWA2 and aPY_." -  this statement is not true for AWA2 I suppose?

5) "_We suspect that pretraining ZSL-KG class representations on ILSVRC 2012 classes help the performance_." How is this pretraining carried out? No mention of this is made in the text or experiment setup.

---

> ### Author Response · Authors · 2022-06-01
> **Clarifications on ZSL-KG**
>
> Thank you for your thoughtful feedback on our work.
>
> ### Requested changes and clarifications.
>
> **##** *It is not clearly explained how or why the class embedding learned by different nodes in the graph would be different from GLoVE/Word2Vec embedding which also model the relationships between words from natural text corpora. A comparison by just using glove (that is not training the graph neural network and just using the initial embedding as is) will make this more clear.*
>
> Thank you for this suggestion. Prior work [1,2] and ours show that graph convolutional networks significantly outperform word vector-based methods. Prior work has compared DeViSE, a method that uses word vectors learned from text corpora, with GCNs (GCNZ, SGCN, DGP) [1,2]. They show that GCNs significantly outperform DeViSE on ImageNet. Our work further improves the performance of GCNs. Additionally, In section 5.1, we compare ZSL-KG with OTyper on the fine-grained entity typing task. OTyper uses GloVe as their class representation. We show that ZSL-KG outperforms OTyper on both datasets.
>
>
> **##** *The performance is almost on par with using GAT on most datasets. Additionally, GAT has time complexity O(N.) for N nodes and m neighbors, while current method is O(N.
> ), so a more detailed analysis on why current method is better than a more simpler GAT is missing. While Table 9 indicates multiple heads might hurt accuracy for GAT, a more accurate comparison should be made using multiple heads for current methods as well.*
>
> We agree that TrGCN is more computationally expensive than GAT but uses a non-linear aggregator. Learning expressive representations using a more sophisticated aggregator opens up a whole new design space for graph neural networks. We include the comparison of ZSL-KG with TrGCN and GAT with multiple heads in Table 9 to show that adding more heads to ZSL-KG-GAT hurts performance. This result suggests that the non-linear aggregator in TrGCN contributes to the difference in performance.
>
> **##** *If two nodes (words) have same neighbors, then according to your formulation, they have exactly the same representation. Firstly, is this possible? If so, is it a good or bad thing for the current problem?*
>
> Thank you for your interesting observation. If two nodes have the same neighbours, then they may not have the same representation because of the influence of the initial features and the graph. During the neighbourhood aggregation, we add a self-loop with the neighbourhood which will result in different representations if they are not the same node. This is true because the aggregation is influenced by the initial node features. If they have the same initial features, then they would have the same representations. On the other hand, TrGCN is limited by some of the issues inherited from GNNs in general. For example, the message passing framework will have the same node representation for all the nodes in a cycle of length 3. Addressing these limitations is ongoing research in the community [3,4,5].
>
> **##** *Sec 5.3 - "Table 5 shows that ZSL-KG, on average, achieves a significantly higher harmonic mean than the graph-based methods on AWA2 and aPY." - this statement is not true for AWA2 I suppose?*
>
> Thank you for pointing out this typo in the manuscript. We are saying that the average of the harmonic mean of ZSL-KG on both AWA2 and aPY is higher than the average performance of the other graph-based methods. We have clarified this in the manuscript (see section 5.3).
>
> **##** *"We suspect that pretraining ZSL-KG class representations on ILSVRC 2012 classes help the performance." How is this pretraining carried out? No mention of this is made in the text or experiment setup.*
>
> We’d like to clarify that we have included the training details in our manuscript’s appendix. For more details see Appendix H.2.
>
> **References**
>
> [1] Wang et.al., Zero-shot Recognition via Semantic Embeddings and Knowledge Graphs. CVPR 2018
>
> [2] Kampffmeyer, Michael, et al. "Rethinking knowledge graph propagation for zero-shot learning." Proceedings of the IEEE/CVF Conference on Computer Vision and Pattern Recognition. 2019.
>
> [3] Hu et.al., How Powerful are Graph Neural Networks? ICLR 2019.
>
> [4] Morris et.al., Weisfeiler and Leman Go Neural: Higher-Order Graph Neural Networks. AAAI 2020.
>
> [5] Chen et. al., Can Graph Neural Networks Count Substructures? NeurIPS 2020.

---

### Review · Reviewer_cjHE · 2022-05-21

**Summary Of Contributions:**

This paper proposed transformer GCN (TrGCN) to address the problem of zero shot learning, or specifically zero-shot classification. The idea is to rely on the knowledge graph (KG) to generate class attributes that are later used for zero-shot classification; this is in contrast to prior works that use word embeddings or a curated graph of words, e.g., WordNet (though to me, KG is already quite close to WordNet as they are both graphs). TrGCN maps nodes in the common sense KG to class attributes (representations) for zero-shot classification. Experiments show that the proposed TrGCN achieve performance that either surpasses or comparable to some existing methods.

**Requested Changes:**

Given KG and WordNet are somewhat different sources of information, it would be interesting to see if they are complementary, that is whether combining them (e.g., DGP and ZSL-KG) would lead to further performance improvement.

While the experiments show some promise, the baselines used seem a bit outdated. The latest baseline for all three tasks is from 2019. It is therefore unclear that the claim that the proposed method achieves new state-of-the-art accuracies is true.


**Strengths And Weaknesses:**

The related work and background sections of the paper is extensive and I believe it would benefit readers with less expertise on this topic. The paper is well organized and the quality of presentation is above the bar for TMLR.

It is also good to see that the experiments cover a diverse range of zero-shot classification applications including Fine-Grained Entity Typing, Intent Classification, and Object Classification.

I appreciate that the authors being upfront about the commonalities and differences between the proposed work and related work including GCGZ, DGP, and GAT. I agree that ZSL with KG as opposed to WordNet introduces new challenges such as the much larger number of nodes and the lack of directions in the graph. However, the difference between GAT and the propose TrGCN is somehow more subtle. It is good to see that empirically TrGCN can outperform GAT.

The proposed method is somewhat straightforward, simple, but seems effective. Given that the guidelines for TMLR focus more on correctness than on impact/significance, I would say that this paper is above the bar for TMLR.

The exploration of using KG as a means to learn class representations for zero-shot classification is useful and the conclusions seem solid and convincing.

Given KG and WordNet are somewhat different sources of information, it would be interesting to see if they are complementary, that is whether combining them (e.g., DGP and ZSL-KG) would lead to further performance improvement.

While the experiments show some promise, the baselines used seem a bit outdated. The latest baseline for all three tasks is from 2019. It is therefore unclear that the claim that the proposed method achieves new state-of-the-art accuracies is true.

---

> ### Author Response · Authors · 2022-06-01
> **Clarifications on ZSL-KG**
>
> Thank you for your thoughtful feedback on our work.
>
> ### Requested changes and clarifications.
>
> **##** *Given KG and WordNet are somewhat different sources of information, it would be interesting to see if they are complementary, that is whether combining them (e.g., DGP and ZSL-KG) would lead to further performance improvement.*
>
> Thank you for raising this interesting point. Combining information from multiple graphs is an important open area in graph neural networks. We are hopeful that exploring larger capacity networks like TrGCN can help with this direction in future work. The question of combining, for example, DGP and ZSL-KG is particularly interesting because WordNet is a subgraph of ConceptNet. However, DGP forms a more complicated GNN structure using the hierarchy in WordNet. Perhaps a kind of GNN that adjusts the message passing structure (or passes multiple types of messages) based on structure within subgraphs would be helpful here.
>
> **##** *While the experiments show some promise, the baselines used seem a bit outdated. The latest baseline for all three tasks is from 2019. It is therefore unclear that the claim that the proposed method achieves new state-of-the-art accuracies is true.*
>
> We would like to point out that we have included comparisons to newer baselines including papers that will be appearing in CVPR 2022 in the appendix (See Appendix A, Table 10). Our results in Table 6 are still consistent with the results in Table 10 and we still achieve the state-of-the-art on aPY dataset. Since we focus on the comparisons with the best-performing methods, we have included extended results in Table 10 of the appendix.

---

> ### Comment · Reviewer_cjHE · 2022-06-07
> **Thanks for the clarification**
>
> Thanks for the clarification, which addresses my main concerns. I would suggest moving some results, especially comparison with more recent baselines (e.g., the one from CVPR'22) to the main text, with one or two sentences clarifying that these baselines, while more recent, underperform SGCN/DGP/GCNZ. This would also strengthen the manuscript.

---

> > ### Author Response · Authors · 2022-06-11
> > **Thank you for comments**
> >
> > We really appreciate your thoughtful comments on our work. As suggested, we will include a discussion with more recent papers in the main body of our manuscript.

---

### Review · Reviewer_C1Vc · 2022-05-23

**Summary Of Contributions:**

The tackles zero-shot learning (ZSL) by exploiting common-sense knowledge graphs. Specifically, the article proposes a new graph neural network, TrGCN, that uses a transformer to aggregate the information of multiple nodes in the graph. The final aggregated information is compared with the features extracted from the input through a dot product. Experiments show that ZSL-KG achieves state-of-the-art results in multiple benchmarks, generally outperforming existing graph-based methods for ZSL.

**Broader Impact Concerns:**

None.

**Requested Changes:**

Following on the above weaknesses, I deem it critical the acceptance that the article:
1. Revises existing claims on the efficiency of TrGCN.
2. Discusses the weaknesses of the proposed approach, especially in the light of Table 7.
3. Expands the related works with existing methods merging GCN and Transformers.
4. Provides deeper evidence on the impact of the knowledge base on the final performance and whether the proposed method is the best in exploiting such a source of knowledge.

To strengthen the work, it would be helpful to show and discuss the results of the model on fine-grained object classification.

**Strengths And Weaknesses:**

I find the idea of deploying a graph neural network on top of ConceptNet for ZSL very interesting. However, some claims of the article are not well supported by experimental evidence and/or algorithmic choices, thus I lean toward a negative rating for the current version of the manuscript. Below I detail what I deem as strengths and weaknesses of the work.

**Strengths:**

1. ZSL models generally assume that unseen classes are described in the same way as seen ones. Collecting precise attribute-level information for all categories can be expensive, especially with a large number of unseen classes. Using knowledge bases for ZSL is a good direction to reduce the burden of category-level attribute annotation, potentially leading to better results than word embeddings.

2. The proposed TrGCN is sound, using a transformer to merge node information in a permutation invariant manner.

3.  The approach achieves good results on a variety of benchmarks (i.e. almost 10% top-5 accuracy on unseen categories for ImageNet, Table 4), being either on par or superior to existing graph-based approaches.

**Weaknesses:**
1. The main weakness of the manuscript is that some claims are not well supported. Specifically:
- Section 4 (second paragraph) underlines how encoding available knowledge graphs with existing models is computationally expensive. In the third paragraph, the text states that the proposed TrGCN addresses this problem. However, it is unclear how TrGCN reduces the computational cost given that i) it has a computational cost that is the power of two of the GAT models, and ii) neighborhood sampling is needed to reduce the computational cost during training (following previous works, e.g. Ying et al. 2018).  These claims should be either reduced or the current limitations of the model explicitly discussed. Note also that, given the focus of the manuscript on using knowledge bases (as for the title), it would be interesting to revise TrGCN in such a way to be computationally more efficient than existing models. It would also be helpful to add an analysis of the computational cost of various models, to assess which one is most viable for working with such large knowledge graphs.

- Table 7 shows that, in some settings, existing GCN and GAT, outperform the proposed TrGCN even on the very challenging ImageNet benchmark. Despite these findings, all the discussion in Section 6 focuses on the advantage of the approach, and further ablation studies (Table 8 and Table 9) are conducted in the two settings (AWA2 and aPY) where TrGCN outperforms the competitors. Note also that Table 9 focuses on the comparison with GAT but does not show how GAT would perform on ImageNet, where it already outperforms TrGCN (according to Table 7). Not discussing the limitations of the approach or its possible drawbacks makes the content of the manuscript not complete and less informative for future works. I would like to stress that if the approach is weaker than the competitors in some settings it is fine, as far as the article properly acknowledges that and discusses the possible reasons, making them valuable starting points for future works.

- In Section 2, the text highlights how existing large-scale models (e.g. CLIP) struggle in fine-grained recognition, while the proposed approach offers a flexible way to encode fine-grained categories. However, the experiments on object recognition are conducted in coarse-grained settings (i.e. AWA2, ImageNet, aPY). Showing results on common fine-grained datasets (e.g. CUB, Flowers, see for instance Xian et al. 2018b) would verify this claim. On the other hand, if case the model struggles in such a setting, the results would nevertheless be interesting for a discussion on the limitations of the approach.

2. The article proposes to couple graph neural networks with transformers, with the TrGCN module. However, this is not the first attempt in the literature, as explored already in e.g. [a,b,c]. In Section 2, there is no discussion on how TrGCN relates to existing works in the same direction. This discussion should be added to stress the technical contribution, clarifying how TrGCN differs from existing works.

3. The title highlights the impact of common sense knowledge graphs. However, the manuscript does not contain any analyses on how the performance of the model varies w.r.t. the used knowledge base (e.g. what if ConceptNet is replaced by WordNet? How would the results of ZSL-KG vary?). This type of analysis would clarify the impact of the knowledge graph w.r.t. one of the proposed technical components, and how sensible the latter is to the underline graph.

4. Following on the previous, it would be helpful to see how the performance of the model compares with existing ZSL models when paired with commonsense knowledge. ConceptNet provides related word embeddings (i.e. ConceptNet Numberbatch). How does ZSL-KG compare with any of the approaches in Table 6 if coupled with such embeddings? And what happens if DGP and SGCN use ConceptNet as a graph rather than WordNet (Section 5.4)? This would allow assessing whether the proposed approach is the best in exploiting commonsense knowledge or whether existing ones can already suffice.

**Minors:**
- $y_n$ denotes a sample in the training set in Section 3, but it is also used to denote an unseen sample and a test sample. I would re-consider writing a more generic $y_i$ when referring to it.
- The set $Y_{U+S}$ is not defined.

**References:**

[a] Yun, Seongjun, et al. "Graph transformer networks." NeurIPS 2019.

[b] Ying, Chengxuan, et al. "Do Transformers Really Perform Badly for Graph Representation?."  NeurIPS 2021.

[c] Hu, Ziniu, et al. "Heterogeneous graph transformer." The Web Conference 2020.

---

> ### Author Response · Authors · 2022-06-01
> **Clarifications on ZSL-KG (1/2)**
>
> Thank you for your thoughtful comments. We have updated the manuscript and addressed your requested changes.
>
> ### Requested changes.
> **##** *Revises existing claims on the efficiency of TrGCN.*
>
> We have updated the manuscript to better explain the efficiency of TrGCN compared to existing graph neural networks (see section 4). We’d like to clarify that it’s important to distinguish between transductive and inductive graph neural networks. Prior work on graph-based zero-shot learning has focused on using transductive methods. Our comments about improved scalability are with respect to these transductive methods, like GCNZ, that require the full graph Laplacian to be known during training. This is computationally impractical for large graphs as GCNZ computes the graph Fourier transform that requires multiplication of the node features with the eigenvector matrix of the graph Laplacian, which is a $O(N^{2})$ for a graph with $N$ nodes. Further, computing the eigenvector matrix is expensive by itself [f]. On the other hand,
> TrGCN is a spatial graph convolutional network that learns node representations by aggregating the local neighbourhood, i.e. TrGCN is inductive. This makes TrGCN computationally practical for large graphs. When TrGCN is compared with other inductive methods like GAT, TrGCN is indeed more computationally expensive because of the transformer that models each neighborhood in the graph. We have already discussed this difference in complexity (see section 4, Differences between TrGCN and GAT).
>
> **##** *Discusses the weaknesses of the proposed approach, especially in the light of Table 7.*
>
> As suggested, we have added a point addressing the weaknesses of TrGCN compared to GAT and GCN (see section 6 paragraph 2). We point out that GAT and GCN outperform TrGCN on BBN and ImageNet datasets. We have also updated Table 8 with the requested additional results on ImageNet. We observe that TrGCN performs the best on AWA2 and aPY, but we find that GAT performs the best on ImageNet (see section 6 paragraph 3). We note that all of these variants of ZSL-KG outperform GZSL, SGCN, and DGP, showing that our choice to emphasize richer knowledge graphs like ConceptNet is also a useful contribution.
>
> **##** *Expands the related works with existing methods merging GCN and Transformers.*
>
> We’d like to clarify that the manuscript does contain comparisons with methods merging GCN and transformers in the related works (see section 2 paragraph 3).  We cite [b], a more recent work that cites and discusses both [a] and [c]. For completeness, we also cite [a,c] in our updated manuscript.
>
> **##** *Provides deeper evidence on the impact of the knowledge base on the final performance and whether the proposed method is the best in exploiting such a source of knowledge.*
>
> We’d like to clarify that the manuscript does contain experiments related to the choice of the knowledge graph. In section 6, we compare SGCN and ZSL-KG-GCN, as they use the same linear aggregator to learn the class representation but train with different knowledge graphs, i.e. SGCN uses WordNet and ZSL-KG-GCN uses ConceptNet.  We see that ZSL-KG-GCN trained on the common sense knowledge graph improves by an average of 6.7 accuracy points across tasks, suggesting that the choice of knowledge graphs is crucial for downstream performance. We have also updated our manuscript to highlight the comparison between the WordNet and ConceptNet graphs (see section 6, Comparison of WordNet and ConceptNet). As suggested in the review, we also compare existing attribute-based methods from Table 6 with ConceptNet embeddings. Our results show that methods that use the graph structure explicitly perform the best on the aPY dataset (see our comment on **##** *Following on the previous, it would…* in the reply).
>
> ### Additional Clarifications
> We would also like to address additional comments in the review.
>
> **##** *Note also that, given the focus of the manuscript on using knowledge bases (as for the title), it would be interesting to revise TrGCN in such a way to be computationally more efficient than existing models. It would also be helpful to add an analysis of the computational cost of various models, to assess which one is most viable for working with such large knowledge graphs.*
>
> Thank you for this suggestion. Recent work in transformers such as Reformer, Informer, Performer, Longformer, etc. show improvements in computational efficiency compared to vanilla transformers. The transformer aggregator in TrGCN can be easily replaced with any of these efficient transformer architectures and make TrGCN computationally more efficient. This would be an interesting direction for future work.

---

> > ### Author Response · Authors · 2022-06-01
> > **Clarifications on ZSL-KG (2/2)**
> >
> > **##** *In Section 2, the text highlights how existing large-scale models (e.g. CLIP) struggle in fine-grained recognition, while the proposed approach offers a flexible way to encode fine-grained categories. However, the experiments on object recognition are conducted in coarse-grained settings (i.e. AWA2, ImageNet, aPY). Showing results on common fine-grained datasets (e.g. CUB, Flowers, see for instance Xian et al. 2018b) would verify this claim. On the other hand, if case the model struggles in such a setting, the results would nevertheless be interesting for a discussion on the limitations of the approach.*
> >
> > Thank you for your suggestion. We would like to point out that our manuscript does have experiments related to fine-grained classification. In section 5.1, we show experiments on fine-grained entity typing datasets. The datasets contain coarse-grained classes such as /person as well as fine-grained classes such as /person/lawyer, /person/politician, etc. Our results show that ZSL-KG performs better than existing methods on the task.
> >
> > ZSL-KG requires the classes to be mapped to the knowledge graph. Fine-grained object classification datasets may contain classes that might be missing from the off-the-shelf ConceptNet graph. For example, ConceptNet does not have a node for the class Forster's Tern in the CUB dataset. To extend ZSL-KG to rare and domain-specific classes, we will need a specialized knowledge graph. We have updated our related work section (see section 2 paragraph 2) to reflect this limitation of ZSL-KG and ConceptNet.
> >
> >
> > **##** *Following on the previous, it would be helpful to see how the performance of the model compares with existing ZSL models when paired with commonsense knowledge. ConceptNet provides related word embeddings (i.e. ConceptNet Numberbatch). How does ZSL-KG compare with any of the approaches in Table 6 if coupled with such embeddings?*
> >
> > Thank you for suggesting this ablation. Recent work has generally shown that using explicit graph structure helps more than the initial embeddings [d]. We show the same by incorporating ConceptNet Numberbatch in DPPN [e. We use DPPN as it is the best-performing attribute-based method on the aPY dataset. DPPN, like ZSL-KG, is a non-generative framework but uses hand-crafted attributes as class representations. We use the DPPN authors’ code to reproduce and train the models. Then, we run another experiment by replacing the hand-crafted attributes with ConceptNet Numberbatch embeddings for the class representations. Our results on aPY dataset suggest that DPPN with ConceptNet has a slightly higher harmonic mean than DPPN with attributes. However, we see that both DGP and ZSL-KG that use graph convolutional networks with knowledge graphs outperform DPPN-ConceptNet. This suggests that the graph neural networks significantly improve the downstream zero-shot performance.
> >
> > |                 | APY  |      |      |
> > |-----------------|------|------|------|
> > |                 | U    | S    | H    |
> > | DPPN            | 40.6 | 55.8 | 46.9 |
> > | DPPN-ConceptNet | 38.1 | 61.7 | 47.1 |
> > | DGP             | 46.2 | 70.2 | 55.7 |
> > | ZSL-KG          | 55.2 | 69.7 | 61.6 |
> >
> >
> > **##** *And what happens if DGP and SGCN use ConceptNet as a graph rather than WordNet (Section 5.4)? This would allow assessing whether the proposed approach is the best in exploiting commonsense knowledge or whether existing ones can already suffice.*
> >
> > We would like to clarify that we have discussed the limitations of DGP that prevent it from being used with ConceptNet. We also compare SGCN with ConceptNet in our manuscript. As pointed out in section 1 (paragraph 4) and section 4 (paragraph 2), DGP requires a directed acyclic graph or parent-child relationship in the graph. In a common sense knowledge graph such as ConceptNet, we are not restricted to a parent-child relationship. Furthermore, we adapt SGCN to ConceptNet, i.e. ZSL-KG-GCN, and show that  ZSL-KG-GCN trained on the common sense knowledge graph improves by an average of 6.7 accuracy points across tasks.
> >
> > **Minor**
> >
> > - **##** *$y_{n}$ ... more generic $y_{i}$ when referring to it.*
> >
> > - **##** *The set $Y_{U+S}$ is not defined.*
> >
> > We have updated our manuscript with your suggestions. We use $y_{i}$ when referring to classes. We also define the set of classes in the generalized setting $Y_{U+S}$.
> >
> >
> > **References:**
> >
> > [a] Yun, Seongjun, et al. "Graph transformer networks." NeurIPS 2019.
> >
> > [b] Ying, Chengxuan, et al. "Do Transformers Really Perform Badly for Graph Representation?." NeurIPS 2021.
> >
> > [c] Hu, Ziniu, et al. "Heterogeneous graph transformer." The Web Conference 2020.
> >
> > [d]  Wang et.al., Zero-shot Recognition via Semantic Embeddings and Knowledge Graphs. CVPR 2018
> >
> > [e] Wang et. al., Dual Progressive Prototype Network for Generalized Zero-Shot Learning, NeurIPS 2021.
> >
> > [f] Kipf et. al. https://tkipf.github.io/graph-convolutional-networks, 2016.
> >
> > [g] Faruqui, M. et. al., Retrofitting word vectors to semantic lexicons. NAACL 2015.

---

> > > ### Comment · Reviewer_C1Vc · 2022-06-07
> > > **Thanks for the reply**
> > >
> > > I thank the authors for their extensive reply, clarifying most of the concerns raised by all reviewers in the first round. I have two concerns left:
> > >
> > > 1. The purpose of Table 7 is to show the advantage of ZSL-KG (with TrGCN) w.r.t. other aggregation strategies. I think that, for completeness, it would be helpful to show if  ZSL-KG is the best only when other types of knowledge bases are used (e.g. WordNet). Note that, even f the proposed approach does not perform well, it would still give more insights into the effectiveness of the model. Moreover, this will allow also the reader to clearly see the impact of ConceptNet on the proposed approach (something that now can only be observed by comparing the results of different tables).
> > >
> > > 2. From the answer to Reviewer MySv, according to Appendix H.2, the model has been pretrained on ImageNet even when applied to AWA2 and aPY. Section 5.3 states that this is done following previous works, however, note that (Kampffmeyer et al., 2019; Wang et al., 2018) do not test on AWA2 and aPY (but only on Imagenet). Is there a particular reason for this choice? In principle, cannot the Glove embeddings be used to initialize the graph nodes and the aggregator function learned on the seen set of the particular dataset? To ensure fair comparisons, the reasons behind it and its eventual impact on the performance should be clear.

---

> > > > ### Author Response · Authors · 2022-06-11
> > > > **Clarifications on ZSL-KG**
> > > >
> > > > Thank you for your thoughtful comments. We really appreciate it.
> > > >
> > > > We would like to address your concerns below:
> > > >
> > > > **##** *The purpose of Table 7 is to show the advantage of ZSL-KG (with TrGCN) w.r.t. other aggregation strategies. I think that, for completeness, it would be helpful to show if ZSL-KG is the best only when other types of knowledge bases are used (e.g. WordNet). Note that, even f the proposed approach does not perform well, it would still give more insights into the effectiveness of the model. Moreover, this will allow also the reader to clearly see the impact of ConceptNet on the proposed approach (something that now can only be observed by comparing the results of different tables).*
> > > >
> > > > As suggested, we compute the results for TrGCN with WordNet and provide insights into the effectiveness of ConceptNet. Our results show that TrGCN with WordNet on OntoNotes outperforms other WordNet-based methods but underperforms the best-performing WordNet-based method on the rest of the datasets. However, we see that TrGCN with ConceptNet, i.e., ZSL-KG always improves the performance compared to TrGCN with WordNet. These inconsistent results suggest that TrGCN can benefit existing applications with WordNet, but might tend to work better with a richer graph structure such as ConceptNet. We will include these results in the Appendix of our paper.
> > > > | Method                      | Ontonotes (Strict) | BBN  (Strict) | SNIPS-NLU  (Acc.) | AWA2  (H) | aPY   (H) |
> > > > |-----------------------------|--------------------|---------------|-------------------|-----------|-----------|
> > > > | GCNZ                        | 41.5               | 21.5          | 82.5              | 73.3      | 58.1      |
> > > > | SGCN                        | 42.6               | 24.9          | 50.3              | 73.7      | 56.8      |
> > > > | DGP                         | 41.1               | 24.0          | 64.4              | 75.1      | 55.7      |
> > > > | TrGCN + WordNet             | 44.4               | 23.1          | 41.9              | 72.2      | 55.4      |
> > > > | ZSL-KG (TrGCN + ConceptNet) | 45.2               | 26.7          | 89.0              | 74.6      | 61.6      |
> > > >
> > > >
> > > > **##** *From the answer to Reviewer MySv, according to Appendix H.2, the model has been pretrained on ImageNet even when applied to AWA2 and aPY. Section 5.3 states that this is done following previous works, however, note that (Kampffmeyer et al., 2019; Wang et al., 2018) do not test on AWA2 and aPY (but only on Imagenet). Is there a particular reason for this choice? In principle, cannot the Glove embeddings be used to initialize the graph nodes and the aggregator function learned on the seen set of the particular dataset? To ensure fair comparisons, the reasons behind it and its eventual impact on the performance should be clear.*
> > > >
> > > > In the results reported in our paper, and in keeping with the literature, all methods use ResNet backbones pretrained on the images in ILSVRC 2012, so in all cases this data is taken as available. The reason that some ZSL methods do not additionally train on the mapping between class descriptions and these images is that the choice of class description determines what training is possible. Methods that use attributes to describe classes are limited because a careful attribute-based description of all 1,000 ILSVRC 2012 classes does not exist. Graph-based methods like GCNZ, SGCN, DGP, and ZSL-KG have an inherent advantage because they use the graph-based class descriptions that are available. We therefore emphasize the comparison among these graph-based methods, which are all trained in the same way, because training without the ImageNet graph information would be an artificial limitation.
> > > >
> > > > We have addressed this concern in the manuscript (see section 5.3> experiment>paragraph 3, and section 5.3>results>paragraph 3). We will update the writing to make it more clear.

---

### Decision · Action_Editors · 2022-07-12

**Recommendation:** Accept with minor revision

**Comment:**

This paper tackles zero-shot learning, where unseen categories must be classified given training only on a dataset with non-overlapping seen categories and, in this case, additional knowledge through as knowledge graphs. The method proposes to use rich knowledge graphs, such as ConceptNet, to learn class representations and the authors propose a transformer-based permutation-invariant non-linear combination function across the neighborhoods. Results are shown across a set of vision and language datasets, including some that are coarse and some that are fine-grained. Overall, the method improves results over the current state-of-art, though not uniformly so.

The reviewers appreciated several aspects of this work, including great writing and structure, the move away from pre-defined attributes and limitations on the knowledge graph (e.g. DAG structures), and results on diverse datasets. However, they pointed out a number of concerns as well, including 1) Increased computational complexity, 2) Fair description of performance claims w.r.t. to similar methods such as GAT (which does better in some cases), and 3) An understanding of how much of the benefits come from richer knowledge graphs (ConceptNet vs. WordNet) or the transformer module. After a thorough rebuttal (including some new results) and discussion with the authors, the reviewers recommended acceptance. Of course the authors should add all revisions coming out of the discussion. Further, it would be valuable to more directly address the computational complexity question in the revision: Specifically, quantitative comparisons of computation and memory would be valuable to the community to clearly show how much is being added to resource requirements; while big O notation is great, there are often large constants or other factors in how the method actually changes these requirements and it should be easy to do such a quantitative characterization.

Overall, assuming such revisions are made, the paper is well-written, provides an interesting update to the use of knowledge graphs, supporting richer structures and proposing an updated transformer-based combination, and does a good job of making reasoned claims and situating the work fairly w.r.t. to prior work. Given the rise of multi-modal models such as CLIP for zero-shot learning, the introduction of richer structures is of interest to the community and this paper does a good job of summarizing this line of work while also proposing a new successful method. Therefore, I recommend acceptance conditioned on revisions.